# Branch point strength controls species-specific *CAMK2B* alternative splicing and regulates LTP

Andreas Franz[1,2] , A Ioana Weber[1] , Marco Preußner[1] , Nicole Dimos[2] , Alexander Stumpf[3] , Yanlong Ji[4,5,6],
Laura Moreno-Velasquez[3], Anne Voigt[3], Frederic Schulz[1] , Alexander Neumann[1], Benno Kuropka[7] , Ralf Kühn[8] ,
Henning Urlaub[4,9], Dietmar Schmitz[3], Markus C Wahl[2,10] , Florian Heyd[1]

**Regulation and functionality of species-specific alternative splicing has remained enigmatic to the present date. Calcium/calmodulin-dependent protein kinase IIβ (CaMKIIβ) is expressed in several splice variants and plays a key role in learning and memory. Here, we identify and characterize several primate-specific *CAMK2B* splice isoforms, which show altered kinetic properties and changes in substrate specificity. Furthermore, we demonstrate that primate-specific *CAMK2B* alternative splicing is achieved through branch point weakening during evolution. We show that reducing branch point and splice site strengths during evolution globally renders constitutive exons alternative, thus providing novel mechanistic insight into *cis*-directed species-specific alternative splicing regulation. Using CRISPR/Cas9, we introduce a weaker, human branch point sequence into the mouse genome, resulting in strongly altered *Camk2b* splicing in the brains of mutant mice. We observe a strong impairment of long-term potentiation in CA3–CA1 synapses of mutant mice, thus connecting branch point–controlled *CAMK2B* alternative splicing with a fundamental function in learning and memory.**

## Introduction

Advances in RNA sequencing have revealed the tremendous impact of alternative splicing on transcriptome diversity, which is especially prevalent in higher order organisms. Alternative splicing is a dynamic process that can be regulated in a tissue-, developmental-, disease-, circadian-, or temperature-dependent manner (Preußner et al, 2014, 2017; Ule & Blencowe, 2019). Similar to gene expression, an extensive network of *cis*-acting sequence elements and associated *trans*-acting protein factors coordinates this process and ensures its fidelity. The basic principles governing splicing regulation have been conserved across evolution, but the complexity of the spliceosome and splicing regulators differs and is likely to generate the regulatory capacity for the vast amount of alternative splicing events observed in some species (Keren et al, 2010; Brooks et al, 2011; Witten & Ule, 2011; Ajith et al, 2016; Ule & Blencowe, 2019). Although several studies have shown that alternative splicing is controlled in a species-specific manner (Graveley, 2008; Barbosa-Morais et al, 2012; Merkin et al, 2012), the regulation and functionality of species-specific alternative splicing remains enigmatic.

It was already revealed during the early stages of the Human Genome Project and similar efforts that the number of protein-coding genes in vertebrates is far below the anticipated number necessary for the observed phenotypic complexity of these organisms. Therefore, early predictions suggested transcriptome diversity generated by alternative splicing to be key in creating biological complexity (Ewing & Green, 2000). In general, the frequency of alternative splicing decreases with evolutionary distance to primates (Kim et al, 2006; Barbosa-Morais et al, 2012) and primate-specific splicing is enriched in frame-preserving events, suggesting functional relevance (Grau-Bové et al, 2018). In addition, alternative splicing patterns have rapidly diverged between species (Modrek & Lee, 2003; Pan et al, 2004), and are now more similar between different organs within one species than they are between the same organs of different species (Barbosa-Morais et al, 2012; Merkin et al, 2012).

Species-specific splicing events appear to be largely *cis*-regulated (Barbosa-Morais et al, 2012; Gao et al, 2015), suggesting that the regulatory principles of *trans*-acting protein factors have largely been conserved during evolution. In the prevailing model,

[1]Freie Universität Berlin, Institute of Chemistry and Biochemistry, Laboratory of RNA Biochemistry, Berlin, Germany   [2]Freie Universität Berlin, Institute of Chemistry and Biochemistry, Laboratory of Structural Biochemistry, Berlin, Germany   [3]Neuroscience Research Centre (NWFZ), Charité - Universitätsmedizin Berlin, Berlin, Germany [4]Bioanalytical Mass Spectrometry Group, Max Planck Institute for Multidisciplinary Sciences, Göttingen, Germany   [5]Hematology/Oncology, Department of Medicine II, Johann Wolfgang Goethe University, Frankfurt am Main, Germany   [6]Frankfurt Cancer Institute, Goethe University, Frankfurt am Main, Germany   [7]Freie Universität Berlin, Mass Spectrometry Core Facility (BioSupraMol), Berlin, Germany   [8]Max Delbrück Center for Molecular Medicine in the Helmholtz Association (MDC), Genome Engineering & Disease Models, Berlin, Germany   [9]Institute of Clinical Chemistry, University Medical Center Göttingen, Göttingen, Germany   [10]Helmholtz-Zentrum Berlin für Materialien und Energie, Macromolecular Crystallography, Berlin, Germany

Correspondence: florian.heyd@fu-berlin.de

species-specific alternative splicing is considered to be the result of a particular combination of binding motifs of splice-regulatory proteins in the vicinity of alternative exons. However, this model falls short of explaining species-specific alternative splicing across different organs with vastly different *trans*-acting environments, leaving the mechanistic basis for species-specific alternative splicing a largely open question.

Few examples of species-specific alternative splicing events have been reported and analyzed in depth. Functional consequences of such events range from altering the activity of RNA-binding proteins (Barbosa-Morais et al, 2012; Gueroussov et al, 2015) to regulating cell cycle arrest (Sohail & Xie, 2015) or converting a noxious heat-sensitive channel into one sensing infrared radiation in vampire bats (Gracheva et al, 2011). In a previous study, we have shown that the strain-specific splicing of *Camk2.1* in the marine midge *Clunio marinus* acts as a mechanism for the natural adaptation of circadian timing (Kaiser et al, 2016). In vertebrates, orthologs of this gene have been identified as key regulators of neuronal plasticity. A potential species-specific regulation could thus have profound repercussions on the acquisition of advanced cognitive abilities in higher mammals.

The calcium/calmodulin-dependent protein kinase II (CaMKII) is a serine/threonine protein kinase that is involved in numerous regulatory pathways (Hell, 2014). In neuronal signaling, CaMKII plays a central role in the integration of the cellular calcium influx, for example, through phosphorylating ion channels, which constitutes a key mechanism underlying synaptic plasticity (Hudmon & Schulman, 2002; Herring & Nicoll, 2016). A unique feature of the kinase is the ability to respond not only to the amplitude but also to the frequency of the activating signal. When the calcium frequency spike exceeds a specific threshold, the enzyme is able to adopt a calcium-independent activation state, which persists even in the absence of the activating signal (Meyer et al, 1992; Chao et al, 2011). This process is considered to be one of the fundamental mechanisms underlying long-term potentiation (LTP), which is widely seen as the molecular basis for learning and memory (Malenka & Bear, 2004).

Whereas organisms such as *Caenorhabditis elegans* or *Drosophila melanogaster* harbor a single ancestral *CAMK2* gene, duplication resulted in a total of four paralogous genes in mammals, termed α, β, γ, and δ (Tombes et al, 2003). These genes and their various splicing isoforms are expressed in a tissue-specific manner, with *CAMK2A* and *CAMK2B* being the predominant isoforms in neuronal cells. Together, they are estimated to constitute up to 1% of total brain protein in rodents (Erondu & Kennedy, 1985) and are by far the most abundant proteins in postsynaptic densities (Cheng et al, 2006). Notably, the conserved features in CaMKII date back to the evolutionary stage when the first synapse is thought to have formed (Ryan & Grant, 2009), and all of its essential features are well conserved among metazoans. Alternative splicing of the four genes leads to the expression of over 70 distinct isoforms in mammals (Tombes et al, 2003; Sloutsky et al, 2020). Genetic variation has mostly been restricted to a variable linker segment that connects the N-terminal kinase domain to a C-terminal hub or association domain. Almost all mammalian splice variants are derived from alternative splicing of one of the nine alternative exons encoding this variable segment. Of the two *CAMK2* genes

predominantly expressed in neurons, *CAMK2A* has three reported alternative splicing isoforms. In contrast, there are 11 known *CAMK2B* isoforms generated by alternative splicing, of which up to eight have been detected in a single tissue (Tombes et al, 2003; Sloutsky et al, 2020). Some of these exons and their respective splice isoforms exhibit tissue- or developmental stage–specific regulation and have been shown to affect the subcellular localization of the enzyme, its substrate specificity, the affinity for the activator calmodulin, and other kinetic properties of the enzyme (Brocke et al, 1995; GuptaRoy et al, 2000; Bayer et al, 2002; O'Leary et al, 2006).

Here, we report species-specific alternative splicing of *CAMK2B* and identify several primate-specific splice isoforms, which are generated through exclusion of exon 16. Minigene splicing assays identify an intronic regulatory sequence responsible for the primate-specific skipping of exon 16. This regulation is independent of the *trans*-acting environment, as primate-specific exon skipping is also observed in mouse cell lines. Using RNA-Seq and minigene analyses, we show that weakening of the branch point (BP) sequence during evolution directs primate-specific exon 16 exclusion. Further systems-wide analyses show that weakening of core *cis*-elements required for splicing, namely, the BP and the splice sites, rendered constitutive exons alternative during evolution. These data add to the mechanistic understanding of how species-specific splicing patterns can be generated independently of the changing *trans*-acting environments of different tissues. Focusing on *CAMK2B*, we show that the primate-specific protein isoforms reach a higher maximal activity in in vitro kinase assays and display different substrate specificities. To address in vivo functionality of *CAMK2B* alternative splicing, we used CRISPR/Cas9 for introducing the human intronic regulatory sequence containing the weaker BP into the mouse genome, which results in strong alteration of *Camk2β* splicing pattern in the brains of mutant mice. Analyses of mice with altered *Camk2β* splicing show strongly reduced LTP in CA3-CA1 hippocampal synapses. As we have not altered exonic coding regions but only intronic splicing-regulatory sequences, this mouse model connects *Camk2β* alternative splicing with neuronal plasticity and suggests a link to species-specific splicing regulation.

# Results

### Alternative splicing of *CAMK2* is species-specific

Alternative splicing of *CAMK2* has long been acknowledged, and multiple studies have reported developmental stage– and tissue-specific splicing events (Tombes et al, 2003; Sloutsky et al, 2020). Differences in splicing between species are known for organisms that are evolutionarily distant from humans, often featuring a single ancestral *CAMK2* gene (Tombes et al, 2003; Kaiser et al, 2016). In vertebrate evolution, *CAMK2* genes have largely been conserved and all mammals harbor the same four genes (Fig 1A). These genes show a conserved architecture, and differences are mostly generated by the presence or absence of certain exons in the variable linker domain. Alternative splicing of *CAMK2* in different vertebrates has been reported, but not systematically compared (Rochlitz et al, 2000; Tombes et al, 2003; Cook et al, 2018; Sloutsky et al, 2020). For a

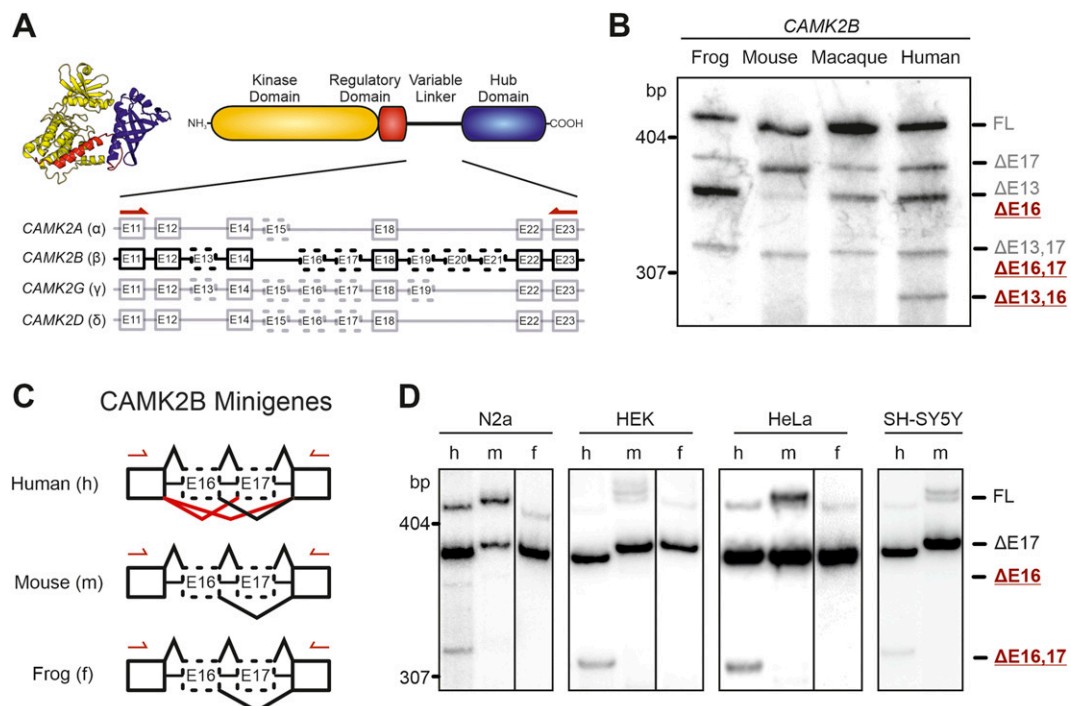

**Figure 1. Species-specific alternative splicing of *CAMK2B* exon 16 is controlled in *cis*.**
**(A)** Schematic representation of the domain architecture of CaMKII and the intron–exon structure of the variable linker region of the four mammalian *CAMK2* genes. Numbered boxes represent exons, and connecting lines represent introns. Boxes with dashed lines represent known alternatively spliced exons. **(B)** Endogenous *CAMK2B* splice isoforms were identified by radioactive isoform–specific RT–PCR with frog (*Xenopus laevis*) and primate (*Macaca mulatta*) total brain RNA, and mouse (*Mus musculus*) and human cerebellum RNA. Isoforms were separated on a denaturing polyacrylamide gel. Isoforms are indicated on the right and named according to the exons that are skipped. As exons 19–21 are missing in neuronal tissue, they were excluded from the naming scheme. **(C, D)** Schematic representation of the minigene constructs used in (D). Red lines indicate primate-specific splicing events. Arrows indicate positions of primer used for RT–PCR. **(D)** Human (h), mouse (m), and frog (f) (*Xenopus laevis*) sequences of exons 16 and 17, including the adjacent introns, were cloned between two constitutive exons and transfected into N2A (mouse), HEK, HeLa, and SH-SY5 (human) cells. Resulting splice isoforms were identified by radioactive RT–PCR. Also see Fig S1.

detailed analysis, we performed radioactive RT–PCR with gene-specific primers on total cerebellum RNA from human and mouse (Fig S1A). Species-specific differences in the alternative splicing pattern can be seen for three of the four *CAMK2* genes (*CAMK2B*, *G*, and *D*; *CAMK2A* shows no difference). For further analyses, we focused on the *CAMK2B* isoform that appears to be exclusively present in human cerebellum.

We extended our analysis to include the rhesus macaque (*Macaca mulatta*) and the African clawed frog (*Xenopus laevis*) (Fig 1B). For both species, the *CAMK2B* splicing pattern resembles that found in mice. All visible splice isoforms were further precisely identified by Sanger sequencing and revealed species-specific alternative splicing of *CAMK2B* exon 16 (previously also named "exon IV/V" [Tombes et al, 2003]), whose inclusion or exclusion leads to three species-specific splice isoforms. The shortest of these, lacking exons 13 and 16 (termed Δ13,16), can easily be identified in the polyacrylamide gel. It is mostly present in humans, but as a faint band is also visible for rhesus macaque, we refer to the exclusion of exon 16 as primate-specific. Of note, RNA-Seq analysis (see below) indicates minor presence of Δ16 isoforms also in mouse cerebellum and potentially other tissues. As we have not observed these isoforms in mice by RT–PCR, we refer to it as primate-specific, although there may be some background

expression in mice as well. Exon 16 is furthermore the least conserved exon in the linker segment, differs in size between the *CAMK2* genes, and, in *CAMK2G*, contains an additional splice donor site (Tombes et al, 2003). It should be noted that exons 19–21 were not present in any of the detected *CAMK2B* isoforms for any of the investigated species. Therefore, the full-length (FL) isoform refers to the longest isoform detected in the cerebellum. Together, these results establish species-specific alternative splicing of *Camk2β*, *γ*, and *δ*, and reveal a novel primate-specific regulation of *CAMK2B* exon 16.

### Species-specific *CAMK2B* alternative splicing is *cis*-regulated

Based on these findings, we designed minigenes for human, mouse (*Mus musculus*, C57BL/6 strain), and frog (*X. laevis*) *CAMK2B*. The minigenes encompass two constitutive *CAMK2B* exons, exon 11 and exon 22, that flank the alternative exons 16 and 17 (Figs 1C and S1B). The introns between exons 16 and 17 and the proximal regions of the flanking introns were included as well. To maintain the intron–exon boundaries of the constitutive exons, the proximal region of their flanking introns was also inserted. The minigenes were transfected into various cell lines and the splicing patterns analyzed by radioactive RT–PCR with a vector-specific primer pair (Fig 1D). The splicing patterns of the minigenes recapitulate the

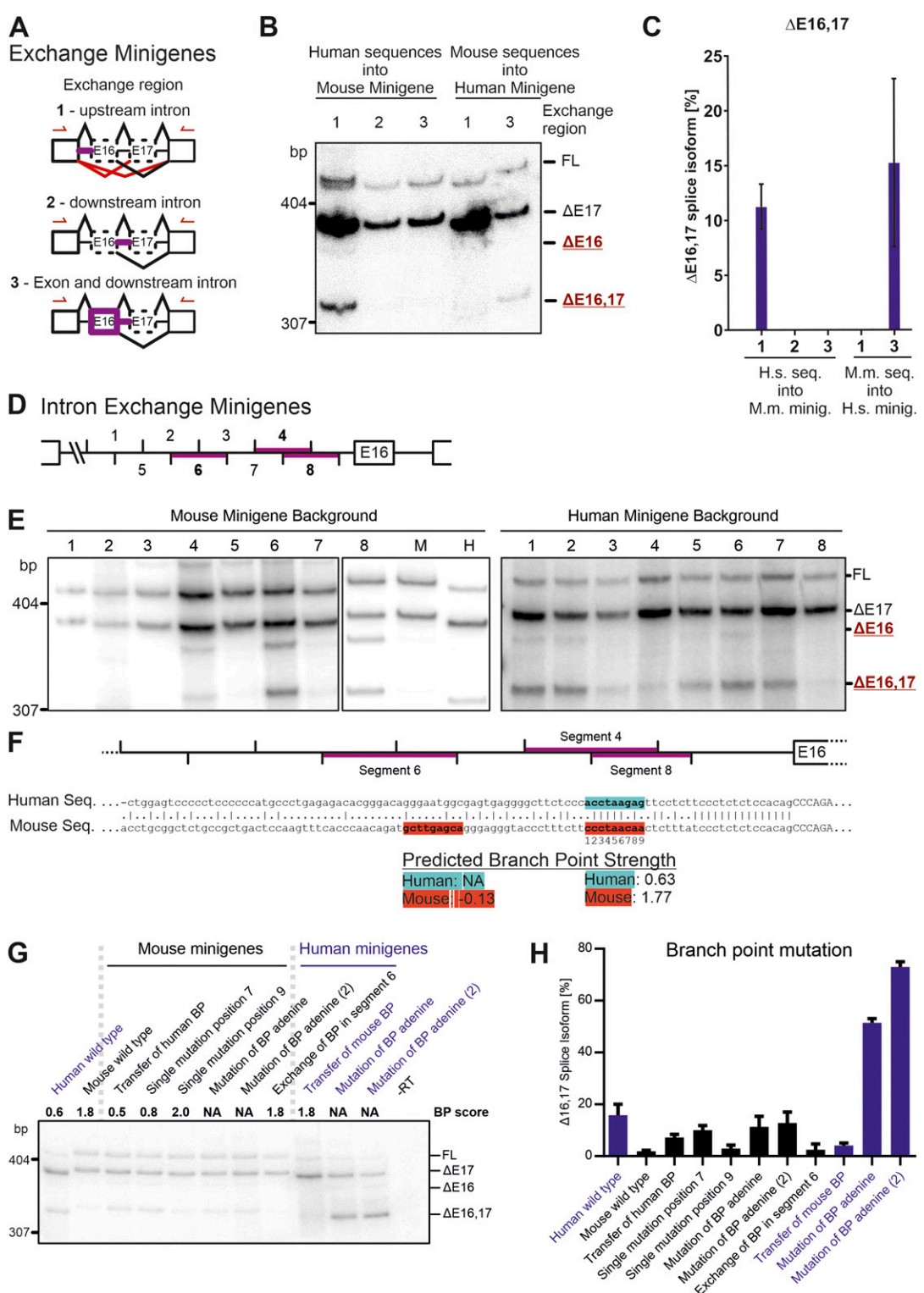

**Figure 2. Branch point strength controls *CAMK2B* exon 16 alternative splicing.**
**(A, B)** Schematic representation of the minigene constructs used in (B). Red lines indicate primate-specific splicing events. Purple lines highlight segments of the minigene that were exchanged between the human and mouse construct. **(B)** Human and mouse exchange minigenes were transfected into HeLa cells and resulting splice isoforms identified by radioactive RT–PCR. **(B, C)** Representative gel (B, C) quantification (n = 3) are shown. **(D)** Schematic representation of the intron containing the identified functionally relevant *cis*-acting element. Numbers indicate 20-bp segments that were exchanged between the human and mouse construct. **(E)** Purple lines highlight segments of functional relevance identified in (E). **(E)** Human and mouse exchange minigenes were transfected into N2A cells and resulting splice isoforms identified by radioactive RT–PCR. **(F)** Sequence alignment between human and mouse of the intron harboring the identified *cis*-acting element. Purple lines highlight

observed endogenous *CAMK2B* splicing patterns. Specifically, all minigenes show bands corresponding to the FL and Δ17 isoforms, whereas only the human minigene shows additional bands for the Δ16 and Δ16,17 isoforms. Transfection of the minigenes into various human and mouse cell lines revealed that the observed splicing pattern is independent of the cell line and species and thus of the *trans*-acting environment. This suggests a *cis*-regulated mechanism, in which differences in the pre-mRNA sequence determine the observed species-specific splicing patterns.

To pinpoint the location of the *cis*-acting element, a second set of minigenes was designed (Fig 2A). In these, intronic or exonic sequences were systematically exchanged between the human and mouse minigenes. Subsequent splicing analyses located the *cis*-acting element to the intron upstream of exon 16 (Fig 2B and C). Insertion of the human sequence into the mouse minigene was sufficient to induce the human splicing pattern. Conversely, transfer of the mouse sequence into the human minigene abolished exon 16 exclusion. Transfer of any other sequence did not lead to an observable change of exon 16 splicing. Together, these observations confirm the primate-specific regulation of *CAMK2B* exon 16 and show that the mechanism is *cis*-regulated, with the regulatory element located in the upstream intron.

### BP strength controls species-specific *CAMK2B* splicing

Having identified the approximate position of the *cis*-regulatory element, we set out to determine its exact location and sequence. As described above, the *CAMK2B* minigenes contain only a part of the intron upstream of the alternative exon 16 (Fig S1B). These 100 bp were further subdivided into eight overlapping segments of 20 bp (Fig 2D). The 3′ splice site itself, including the first 15 bp upstream of it, is identical in humans and mice and was thus not included in the analysis. The eight segments were exchanged between the human and mouse minigenes and the resulting splicing patterns analyzed after expression in cell lines from both species (Figs 2E and S2). No difference between the tested human and mouse cell lines was observed, further supporting the *cis*-regulated nature of the splicing event. RT–PCRs identified three segments of functional importance, two of which overlap by 10 bp. These two segments (segments 4 and 8) acted in both ways and are thus necessary and sufficient: transfer of the mouse sequence into the human minigene was sufficient to abolish the human-specific exclusion of exon 16, whereas transfer of the human sequence into the mouse context induced exclusion of exon 16. The third identified segment (segment 6) only worked in one direction: transfer from human to mouse induced exon 16 exclusion, whereas the corresponding mouse sequence inserted into the human minigene did not change the splicing pattern. Together, these findings reveal two sequences in the intron upstream of *CAMK2B* exon 16 that regulate its species-specific alternative splicing.

Intriguingly, prediction of potential BP sequences (Corvelo et al, 2010; Nazari et al, 2019) revealed that both species harbor the most

salient BP sequences in the overlap of segments 4 and 8 (Fig 2F). Although this sequence resembles a near-optimal BP that lies within the AG dinucleotide exclusion zone (AGEZ) in the mouse intron, the corresponding human sequence scores much lower and lies slightly outside of the AGEZ. Including the other two species for which we have analyzed the endogenous splicing pattern (rhesus macaque and African clawed frog), the predicted BP strength ranks mouse > frog > macaque > human (Table 1) and correlates well with the observed splicing pattern. The predicted frog BP sequence scores lower than the corresponding mouse sequence, but an alternative BP is found in very close proximity, and, when summed, the combined frog BP strength reaches that of the mouse BP. The predicted macaque sequence scores higher than the corresponding human sequence but considerably lower than the mouse sequence. Notably, in the RT–PCR, the rhesus macaque sample also shows a faint band for the Δ13,16 exclusion isoform for endogenous *CAMK2B* (Fig 1C).

We also analyzed the splice site strengths of the orthologous *CAMK2B* exon 16. As expected from the strong conservation of the splice site-proximal nucleotides, the predicted strength of the 3′ splice site does not substantially differ between mouse and human, suggesting it is a consensus splice site in both species (MaxEntScan, human: 12.03; mouse: 13.53 [Yeo & Burge, 2004]). Similarly, the 5′ splice site is strongly conserved between both species (MaxEntScan, human: 4.41; mouse: 4.41). These data suggest that BP evolution has a substantial impact on species-specific alternative splicing, with a suboptimal BP in humans rendering the exon alternative and thus generating additional *CAMK2B* complexity when compared to constitutive inclusion in mouse and frog.

To validate these findings, we designed variants of our established minigene constructs to specifically modify the predicted BP sequences (Fig 2G and H). Exchange of the 9-bp BP motif alone was sufficient to confer species-specific splicing of exon 16 in both directions. Targeted mutation of individual nucleotides revealed that a single C to G mutation at position 7 in the mouse BP motif is sufficient to lower the predicted BP strength and induce primate-specific exon exclusion. Mutation of the BP adenine itself has a similar effect for the mouse minigene, resulting in a splicing pattern reminiscent of the human minigene. The orthogonal mutation in the human minigene has a more drastic effect, leading to 60–80% exclusion of exon 16. This suggests the existence of additional BPs in the mouse minigene, which are absent in the human ortholog. As we had identified exchange segment 6 to be functionally relevant in the mouse sequence (Fig 2F), we exchanged a potential BP in this mouse segment to the human sequence, which does not contain a predictable BP. However, this did not alter splicing regulation, suggesting that this BP has a minor, if any, contribution to controlling exon 16 splicing in mice. Taken together, these results strongly suggest that the *cis*-regulatory element is not an intronic or exonic splicing enhancer or silencer motif, but that the splicing differences are instead mediated by the alteration of core splicing elements, namely, of the BP sequences. Although this conclusion is

segments of functional relevance. Highlighted sequences indicate locations of predicted BPs (Corvelo et al, 2010). **(G)** BP mutation minigenes were transfected into N2A cells and resulting splice isoforms identified by radioactive RT–PCR. **(G, H)** Quantification of (G). Error bars indicate SD, n = 3. Also see Fig S2.

**Table 1. Predicted branch point scores for the intron upstream of *Camk2β* exon 16.**

| Species | AGEZ | Distance | Sequence | Branch point score |
|---|---|---|---|---|
| Mouse | 44 | 26 | ccctaacaa | 1.77 |
| Frog (BP #1) | 18 | 20 | aactaagtc | 1.11 |
| Frog (BP #2) | 18 | 24 | ctttaacta | 0.74 |
| Rhesus macaque | 24 | 26 | gcctaaggg | 0.83 |
| Human | 22 | 26 | acctaagag | 0.63 |

BP scores were calculated using SVM-BP (Corvelo et al, 2010). AGEZ: AG dinucleotide exclusion zone; distance: distance to 3′ splice site; sequence: sequence of identified BP; branch point score: predicted BP score (scaled vector model).

entirely consistent with our minigene analyses, additional *cis*-acting elements outside the sequence included in our minigene may contribute to the regulation in the endogenous situation.

### A weak BP correlates with *CAMK2B* exon 16 skipping across primates

To confirm these findings, publicly available RNA-Seq data from different mammals were analyzed, with a focus on primates. RNA-Seq data from cerebellar tissue of different species or, where cerebellum data were not available, total brain tissue, were mapped to the corresponding genomes (Fig 3A). Exon 16 and exon 16,17 exclusion isoforms could be confirmed in humans, even though they only amount to ~5–7% of all *CAMK2B* transcripts. Exclusion of exon 16 was also observed in mouse tissue, but at the much lower frequency of ~0.4%. Even less exon 16 skipping was observed in the evolutionarily more distant pig (*Sus scrofa*), whereas all analyzed primates show substantial exon 16 skipping. Alignment of the BP sequences showed a very high degree of similarity between all primates and several differences between these species and mouse or pig (Fig 3B). The latter two species show a significantly higher BP strength, which correlates with the observed splicing pattern differences, as compared to primates. The core of the BP motif seems to be conserved among primates, with only two nucleotides showing some variation. These variations correlate with the evolutionary relationship and result in slightly different predicted BP strengths. By associating exon 16 exclusion levels and BP strength, we observe two distinct clusters of either low exon 16 exclusion with a strong BP (mouse, pig) or high exon 16 exclusion with a weak BP (primates) (Fig 3C). We note that even though the exact percent spliced in (PSI) values differ between human, chimpanzee (*Pan troglodytes*), bonobo (*Pan paniscus*), and gorilla (*Gorilla gorilla*), their BP sequences are identical. A recent study analyzed the expression of CaMKII in human hippocampi and found the prevalence of exon 16 exclusion isoforms of *CAMK2B* to range from ~4% to 16% between different tissue donors (Sloutsky et al, 2020). This suggests additional regulatory layers that are specific to individual samples, for example, donor age, developmental stage, or the precise brain region that was used, and may provide an explanation for the differences we observe.

We then extended our analysis regarding the conservation of the BP sequence to include additional species (Fig S3). All primates, and also the order *Dermoptera* (the flying lemurs), the closest living relatives of primates, show a weak BP. All other species harbor a strong BP motif that shows a medium degree of sequence conservation among most mammals. The sequences diverge with increased evolutionary distance, but the high BP strength is maintained. Notable exceptions, such as the lizard (*Anolis carolinensis*), have intron sequences that do not return any valid, predicted BPs in close proximity to the splice site, suggesting fundamental differences in the splicing machinery or consensus BP sequences. Across all analyzed species with low or high BP scores, a BP core sequence is highly conserved: from positions 2 to 6 (Fig S3, CCTAA). It is interesting to note that, aside from this sequence, only the G at position 7 is conserved across all primates, whereas we observe variability across primates in all of the other positions (Fig S3). It is precisely the G at position 7 that, when introduced into the mouse sequence, strongly decreases the BP score and leads to exon exclusion (Fig 2F and G). The finding that the residue crucial to maintain a functional but weak BP is conserved across all primates, while other residues outside of the conserved BP core show variability in different primates, points to a positive selection for the weak BP in primates, which permits partial exon skipping and increases diversity of *CAMK2B* transcripts and proteins. That this crucial residue and the weak BP are maintained throughout primates while other nucleotides in the BP sequences show variations that do not impact on the BP strength, further argues against this splicing event being tolerated noise and suggests that maintaining exon 16 alternative splicing is selected for during evolution. This supports the idea of an important functional role of the respective splice variants.

### BP strength globally controls species-specific alternative splicing

We next addressed whether species-specific differences in the BP motifs globally regulate species-specific alternative splicing. To this end, we first defined orthologous exons between mouse and human (see the Materials and Methods section), and then analyzed RNA-Seq data from a large collection of human and mouse brain samples. This approach allowed us to define orthologous exons that are alternatively spliced in both species or that are alternative exclusively in mouse or human brain (Supplemental Data 1). Consistent with previous reports (Barbosa-Morais et al, 2012; Merkin et al, 2012), we observed a higher number of exons that are alternative only in humans (Figs 4A and B and S4A and B). We then analyzed these species-exclusive subsets of alternative exons and observed clear differences in the strengths of their core splicing elements (Fig 4C and D). Exons that are exclusively alternative in

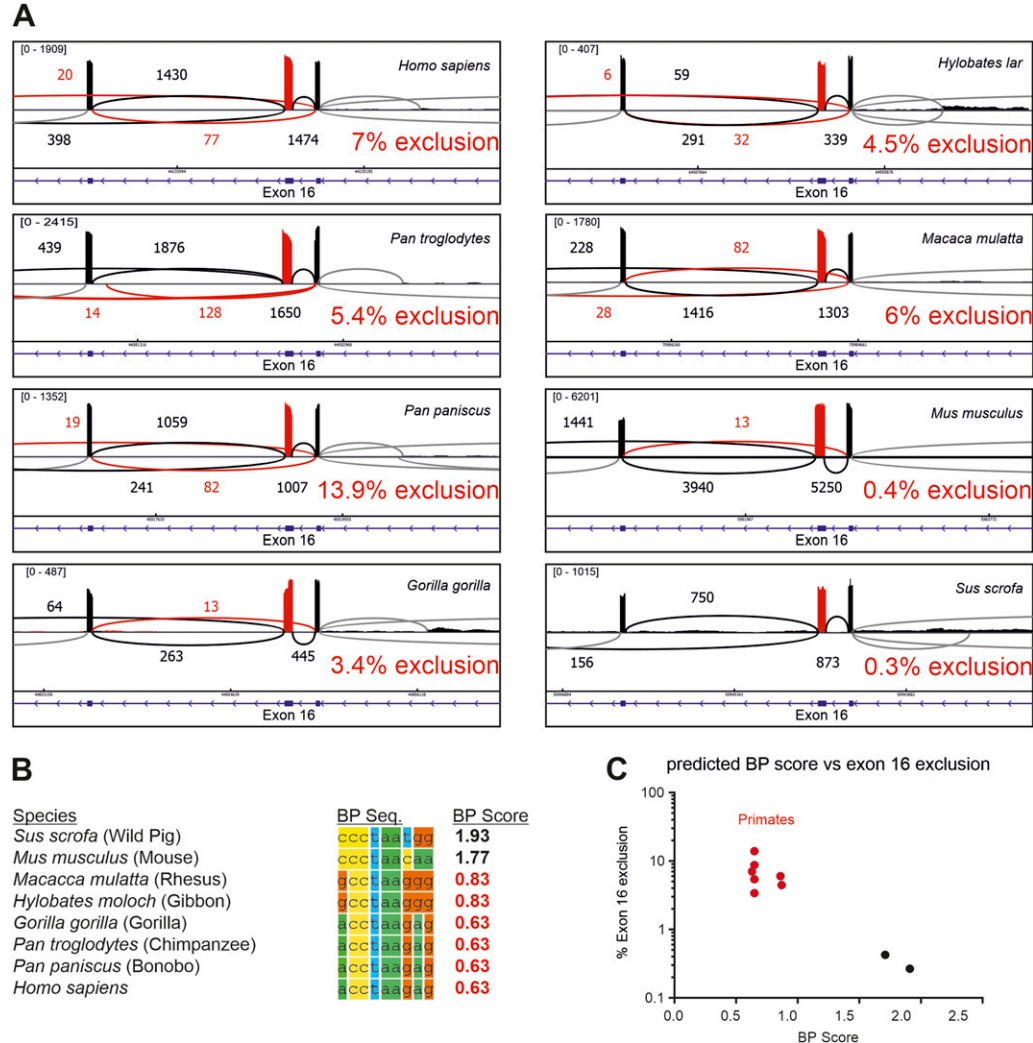

**Figure 3. Evolutionary adaptation of branch point strength controls primate-specific *CAMK2B* exon 16 skipping.**
**(A)** Sashimi plot from STAR showing the alternative splicing of *CAMK2B* exon 16 in RNA-Seq data from human, chimpanzee (*Pan troglodytes*), bonobo (*Pan paniscus*), gorilla (*Gorilla gorilla*), orangutan (*Pongo abelii*), gibbon (*Hylobates lar*), rhesus macaque (*Macaca mulatta*), mouse (*Mus musculus*), and pig (*Sus scrofa*). RNA-Seq data from cerebellum were used for all species, except orangutan, for which RNA-Seq data from total brain tissue were used. Red color indicates exon 16 and exon 16 exclusion reads. Numbers indicate number of reads per splice junction, with the minimum set to 3 junction reads. Shown in blue is the intron–exon structure of the displayed region. % exon 16 exclusion is indicated. Exclusion values were generated from RNA-Seq data using rMATS (Shen et al, 2014). **(B)** Alignment of the identified functionally relevant BP sequence. The BP strength was predicted using SVM-BPfinder (Corvelo et al, 2010) with the human BP model. BP score refers to the BP motif score (scaled vector model). **(B, C)** Predicted BP strength from (B) was plotted against the exon 16 exclusion levels determined by RNA-Seq. Also see Fig S3.

humans show overall weaker BP sequence scores, BP motif scores (the BP motif score includes the distance to the 3' splice site), and 3' and 5' splice site scores (Fig 4C) when compared to their mouse orthologs that are constitutively spliced. A similar trend can be observed for mouse-exclusive alternative exons, which show reduced BP and splice site scores when compared to the constitutive human orthologs (Fig 4D). Importantly, this effect is restricted to the alternatively spliced exon itself, as the core splicing elements of the surrounding constitutive exons do not show any prominent differences between the two species (Fig S4C). These data add to a mechanistic understanding of how global species-specific splicing patterns are established through the evolution of the core splicing sequences. Evolutionary weakening of splice sites, especially the 5' splice site (Lev-Maor et al, 2007; Gelfman et al, 2012), has previously been suggested to contribute to species-specific alternative splicing. We now provide evidence that weakening of the BP sequence plays a similar role in allowing alternative usage of an exon, thus increasing transcriptome and proteome complexity through suboptimal exon recognition. Notably, this model also provides an explanation for species-specific alternative splicing that is at least partially independent of the distinct *trans*-acting environments in different cell types and organs.

### Primate-specific CaMK2β isoforms display slightly increased activity

Having established the genomic causes and the transcriptomic consequences of the species-specific alternative splicing of

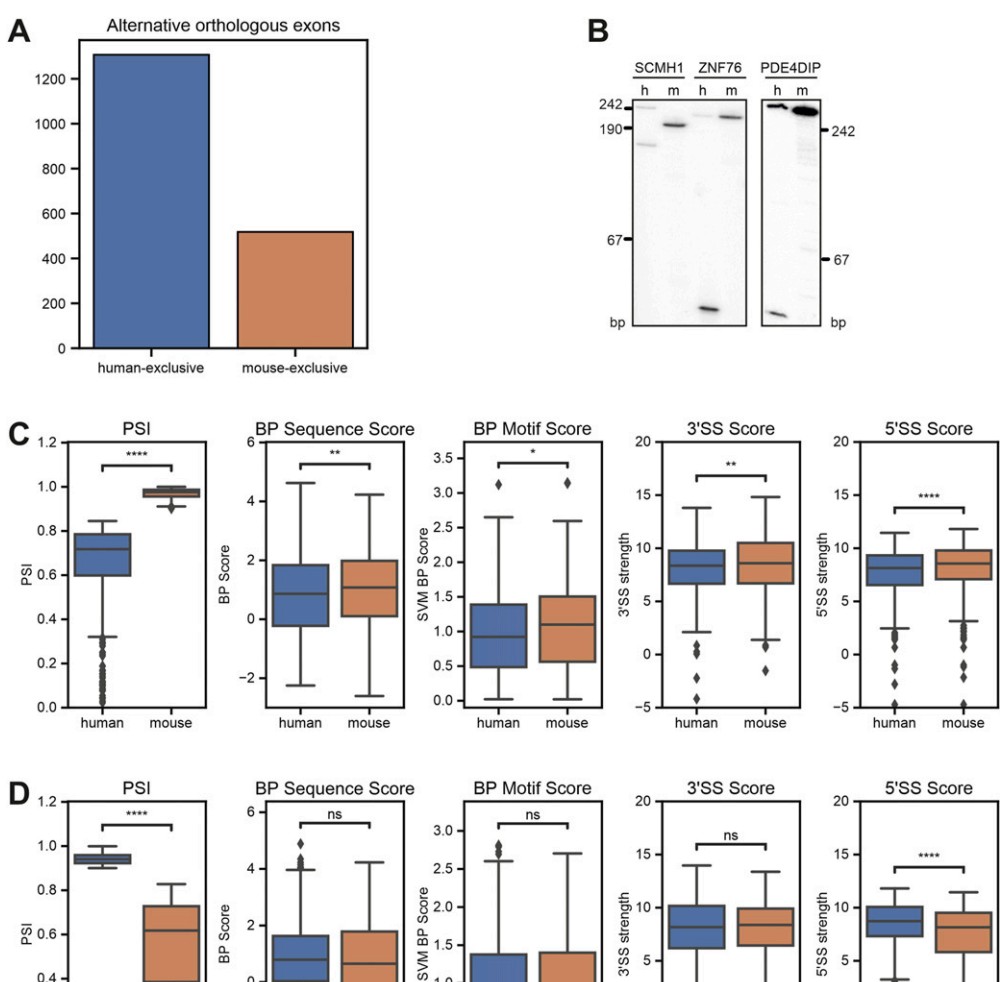

**Figure 4. Branch point and splice site strength globally control species-specific alternative splicing.**
**(A)** Species-exclusive alternative orthologous exons. RNA-Seq data from different brain regions from mouse (n = 47) and human (n = 9) were analyzed to identify species-specific splicing pattern. The analysis was restricted to orthologous exons (see the Materials and Methods section for details) that are alternatively spliced in one species (PSI < 0.9) but not the other (PSI > 0.9) (see Supplemental Data 1). **(B)** Validation of species-exclusive alternative exons by radioactive RT–PCR. m, mouse; h, human. **(C, D)** Boxplots comparing human and mouse splicing element scores for human-exclusive (C) or mouse-exclusive (D) alternative orthologous exons. PSI, percent spliced in; BP Sequence Score, branch point sequence score; BP Motif Score, branch point motif score using a scaled vector model (Corvelo et al, 2010); 3′/5′SS Score, splice site score (Yeo & Burge, 2004). *P < 0.05, **P < 0.01, and ***P < 0.001 (Wilcoxon's signed-rank test).

*CAMK2B*, we set out to determine its effect on the protein level. We selected two species-specific isoforms (Δ16,17 and Δ13,16) and two control isoforms (FL and Δ13) for recombinant production and purification from insect cells (Fig 5A). Again, the FL isoform refers to the longest detected isoform in cerebellum and lacks exons 19–21 (Fig 1A and B). These four isoforms were tested in a radioactive in vitro kinase assay with the model substrate Syntide 2 (Hashimoto & Soderling, 1987), linked to GST (Fig 5B and C). Activity was monitored as a function of calmodulin concentration to test the cooperativity of the enzyme. Consistent with a recent publication (Sloutsky et al, 2020), we did not observe major differences in the $EC_{50}$ values or the Hill coefficients (Table 2) between the four CaMKIIβ variants. Instead, we observed small but significant differences in the maximal activity ($V_{max}$) reached at optimal calmodulin concentrations (Fig 5D). At concentrations of 100–1,000 nM calmodulin, both primate-specific protein isoforms reach a slightly higher maximal activity compared with the FL and Δ13 isoforms. The same effect was also observed using human FL τ protein (τ-441) as an alternative CaMKIIβ substrate (Fig S5A and B).

One of the key properties of CaMKII is its ability to adopt different activation states, based on its own phosphorylation pattern (Bayer & Schulman, 2019). Upon stimulation, the enzyme quickly *trans*-autophosphorylates on T287 and adopts an autoactivated state that persists even in the absence of calcium/calmodulin. Recent studies suggest that the rate at which certain activating and inhibiting phosphorylations are acquired differs between CaMKII protein isoforms and might also be influenced by the length and composition of the variable linker segment (Bhattacharyya et al, 2020). We thus tested the autoactivity of the selected CaMKIIβ isoforms at varying calmodulin concentrations, which directly reflects the activation and hence phosphorylation state of the enzyme. CaMKII was first stimulated in the absence of the substrate protein, after which calcium was quenched by adding EGTA. Addition of the substrate Syntide 2–GST allowed assessment of the previously generated autoactivity. Similar differences in $V_{max}$ could be observed, with the primate-specific protein isoforms reaching slightly higher maximal activities (Fig S5C and D). Together, these results confirm that the tested CaMKIIβ splice isoforms do not differ in their

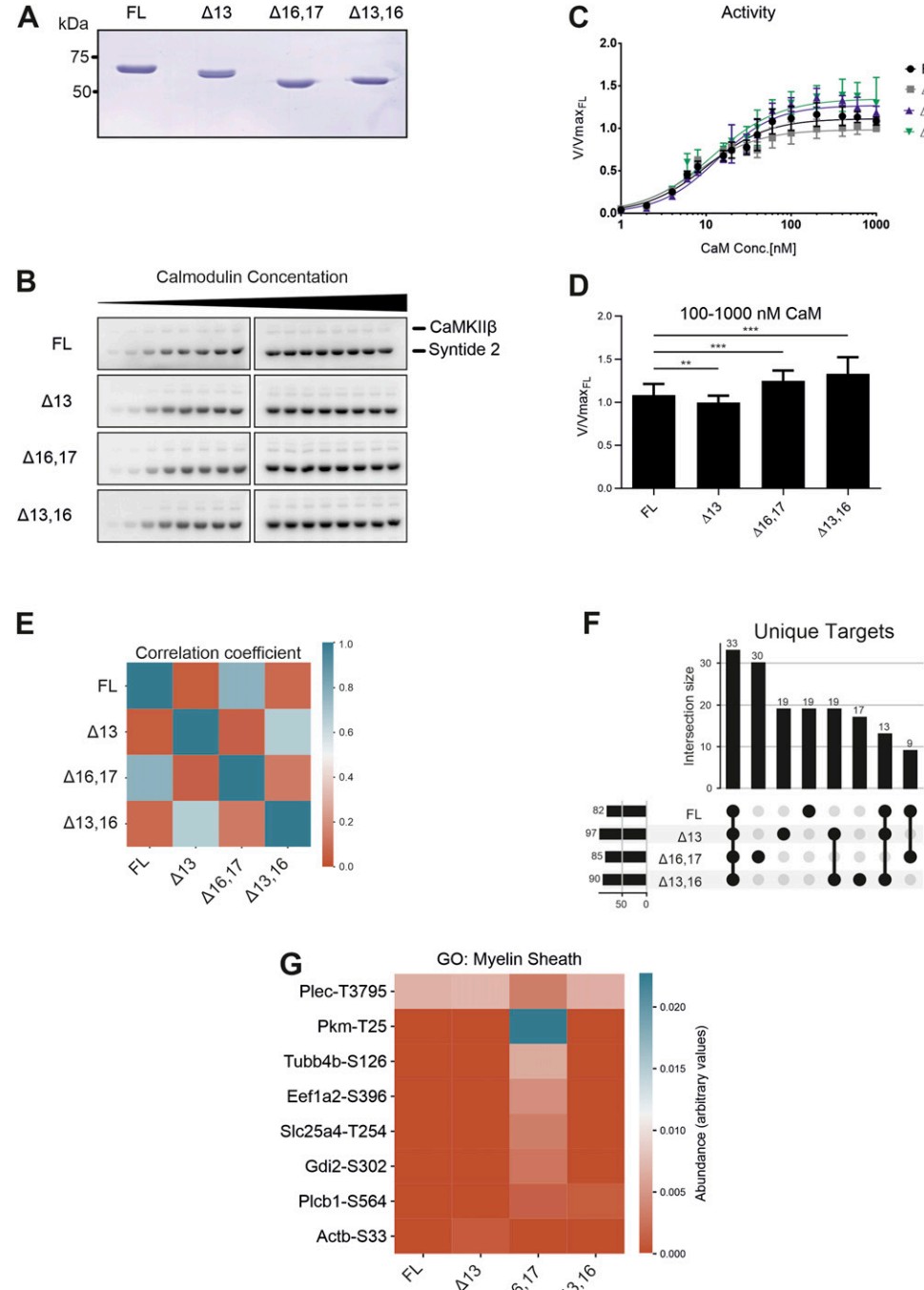

**Figure 5. CaMKIIβ protein isoforms differ in their kinetic properties and substrate spectra.**
**(A)** SDS–PAGE of purified CaMKIIβ isoforms. Proteins were expressed in insect cells and purified via Strep-affinity and size-exclusion chromatography. Protein concentration was determined via UV absorption at 280 nm and precisely leveled by repeated SDS–PAGE, Coomassie staining, and subsequent quantification. **(B)** In vitro kinase assay with different CaMKIIβ isoforms. CaMKII activity against a protein substrate (Syntide 2, fused to GST) was measured as a function of calmodulin concentration. Direct phosphorylation of the substrate by CaMKIIβ was measured via $^{32}$P incorporation. Samples were separated on an SDS–PAGE and detected using autoradiography. **(B, C, D)** Quantification of (B), normalized to the maximum activity of the FL isoform (n = 6). Error bars indicate SD. Data were fitted to a Hill equation. **(D)** Samples at maximal activity were combined. Error bars indicate SD. *$P$ < 0.05, **$P$ < 0.01, and ***$P$ < 0.001 calculated by $t$ test or Welch's $t$ test and adjusted for multiple comparisons using Holm's method. **(E)** Correlation matrix of the substrate spectra of different CaMKIIβ isoforms, as determined by an analog-sensitive kinase assay. The analysis was restricted to CaMKIIβ-specific targets. A Person correlation coefficient was calculated based on the intensity values of individual phosphorylation sites. **(F)** Intersection plot showing the isoform- and group-exclusive phosphorylation sites. Analysis was restricted to CaMKIIβ-specific targets. Numbers on the left indicate the total number of phosphorylation sites detected in a sample. Numbers on the top indicate the intersection size between samples, meaning the number of phosphorylation sites that are unique to this group of samples. Black dots and connecting lines indicate the exact group of samples for which the intersection size is displayed. **(G)** Heatmap showing the abundance of individual phosphorylation sites associated with the GO term "myelin sheath" (GO: 0043209) in the substrate spectra of different CaMKIIβ isoforms. Also see Figs S5 and S6.

**Table 2. Kinetic parameters of purified CaMKIIβ isoforms.**

| Parameter | Substrate: Syntide 2 | | | | Substrate: τ-441 | |
|---|---|---|---|---|---|---|
| | Isoform | | | | Isoform | |
| | FL | Δ13 | Δ16,17 | Δ13,16 | FL | Δ16,17 |
| Vmax | 1.11 ± 0.02 | 0.99 ± 0.02 | 1.27 ± 0.03 | 1.35 ± 0.03 | 1.04 ± 0.02 | 1.25 ± 0.03 |
| h | 1.22 ± 0.11 | 1.20 ± 0.09 | 1.29 ± 0.11 | 1.11 ± 0.11 | 1.53 ± 0.19 | 1.58 ± 0.17 |
| EC50 | 10.30 ± 0.82 | 7.80 ± 0.56 | 13.65 ± 0.97 | 11.98 ± 1.16 | 12.2 ± 0.90 | 18.8 ± 1.30 |

Kinetic parameters as determined by in vitro kinase assay and subsequent fitting of a Hill equation. h, Hill coefficient.

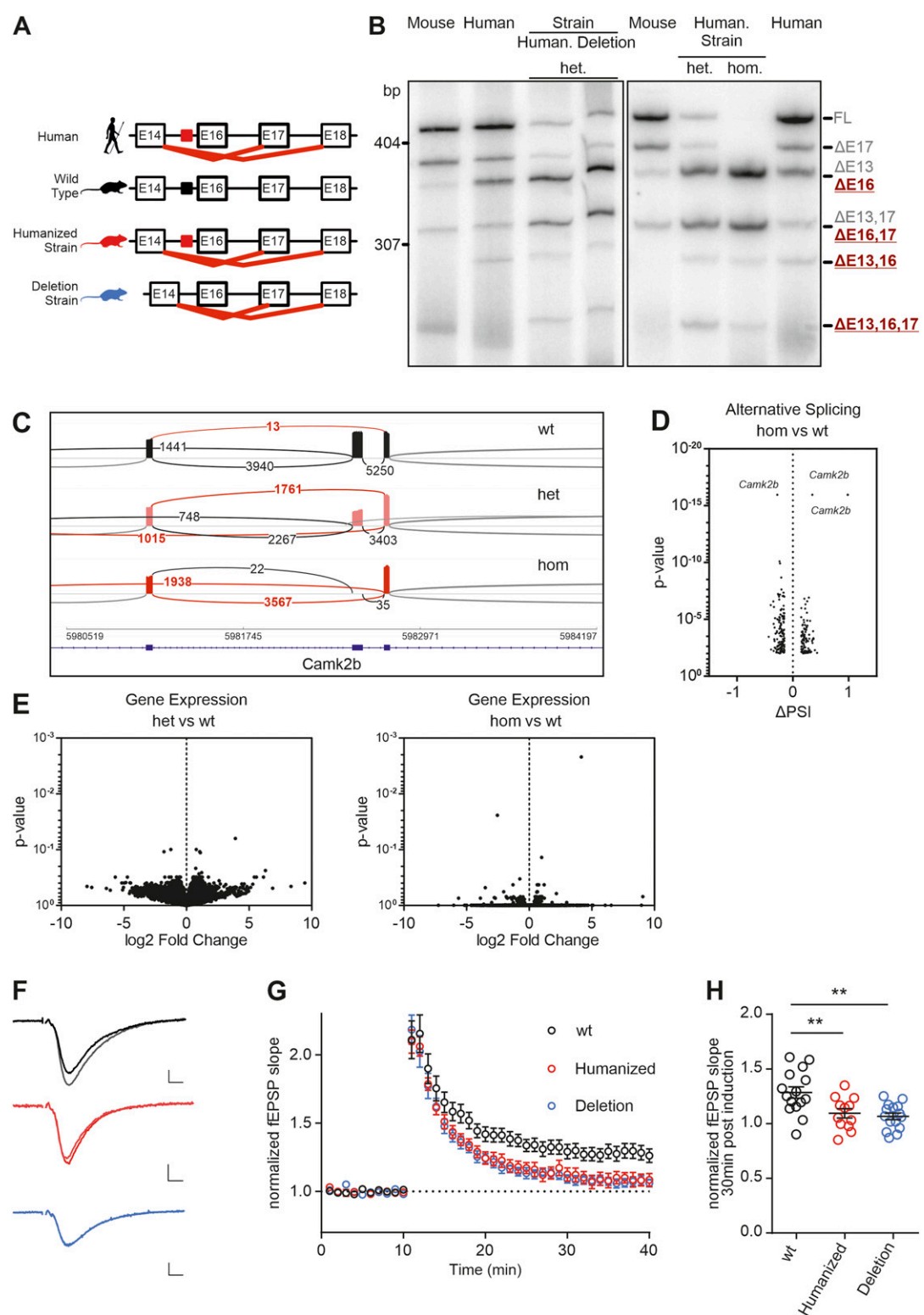

**Figure 6. Mouse model with humanized *Camk2b* exon 16 branch point shows strong impairment in LTP formation.**
**(A)** Schematic representation of the intron–exon structure of the variable linker region of the *CAMK2B* gene and comparison of the identified alternative splicing isoforms in human, WT mice, and the mouse model for *Camk2b* exon 16 skipping (humanized strain and deletion strain). Red lines indicate identified species-specific splicing events. Colored boxes indicate the location of the identified *cis*-regulatory element in human (red) and mice (black). **(B)** Endogenous *Camk2b* splice isoforms were identified by radioactive isoform–specific RT–PCR with mouse and human cerebellum RNA. Isoforms were separated on a denaturing polyacrylamide gel. Isoforms are indicated on the right and named according to the skipped exons. Human, humanized strain; Deletion, deletion strain; WT, wild-type animals; het, heterozygous animals;

EC$_{50}$ values or Hill coefficients. Instead, a slight difference in maximal activity at optimal calcium/calmodulin concentrations can be observed. Notably, this may allow the primate-specific variants to react more strongly to calcium influx and may thus contribute to translate primate-specific alternative splicing into functionality.

## CaMKII$\beta$ isoforms have different substrate spectra

In addition to subtle kinetic variations between the CaMKII$\beta$ isoforms, we considered potential differences in their substrate spectra as a further mechanism for diversified functionality. Instead of testing individual substrates in vitro, which has previously been done for fly CaMKII (GuptaRoy et al, 2000), we employed the analog-sensitive kinase system (Lopez et al, 2014). This approach allows for direct labeling of kinase substrates in complex samples and does not require prior knowledge of potential phosphorylation targets.

An analog-sensitive variant has previously been described for CaMKII$\alpha$ (Wang et al, 2003), and consistent with the high sequence similarity of the kinase domains, the same residue exchange (F89G) was effective in creating a CaMKII$\beta$ variant that could use ATP analogs with bulky side chains on their N$^6$ atoms (Lopez et al, 2014). We confirmed in vitro and in cells that the analog-sensitive variant exhibited similar enzymatic activity as the WT enzyme, that only the variant could be competitively inhibited by bulky ATP analogs, and that, in permeabilized N2A cells, the ATP analog N$^6$-benzyl-ATPγS was used by only the variant kinase (Fig S6A–E).

The four CaMKII$\beta$ isoforms that were analyzed in in vitro kinase assays and two additional control samples—untransfected (UT) and a kinase-dead variant (K43R)—were chosen for kinase assays with N$^6$-benzyl-ATPγS in permeabilized N2A cells, subsequent thio-phosphate enrichment, and detection of substrates via mass spectrometry (MS) analysis (Supplemental Data 2). To identify qualitative and quantitative differences in the substrate spectra, we excluded targets also identified in the control samples and any that mapped to the alternative exons in the variable linker itself. We then generated a correlation matrix based on the abundance of the identified phosphorylation sites. Strong correlations between the FL and Δ16,17 isoforms on the one hand and between the Δ13 and Δ13,16 isoforms on the other were observed, indicating that different splice isoforms have preferred substrates (Fig 5E). Interestingly, the correlation between the FL and Δ16,17 isoforms is mainly based on similar CaMKII autophosphorylation (Fig S6G and Table S1), which likely controls kinase activity and/or localization. Comparing substrates that are phosphorylated by the different variants also identified substrates that are exclusively phosphorylated by individual isoforms, including 17 targets of the primate-specific Δ13,16 isoform. We also note that the largest intersection is

between all four CaMKII$\beta$ isoforms, indicating a relatively large overlap in their substrate spectra (Fig 5F). Although we did not observe clear-cut differences in the gene ontology (GO) terms of isoform-specific substrates, our data suggest that CaMKII$\beta$ isoforms have isoform-preferred/specific substrates. For example, substrates only found for the Δ16,17 isoform are enriched in the GO term "myelin sheath" (GO:0043209) (Fig 5G), suggesting a potential isoform-specific functionality. Interestingly, a substrate specifically phosphorylated by the primate-specific isoforms is the catalytically relevant Y-box of phospholipase C-$\beta$1 (Plcb1) (Supplemental Data 2), an enzyme involved in inositol triphosphate (IP$_3$) signaling that has been connected to learning and memory (Cabana-Domínguez et al, 2021). Although individual targets have not yet been validated, these data suggest that CaMKII$\beta$ isoforms differ in their substrate preferences, with primate-specific isoforms targeting several proteins related to key neuron functions.

## A mouse model for *Camk2b* exon 16 exclusion

To study the consequences of *CAMK2B* alternative splicing in vivo, we generated a mouse model for *Camk2b* exon 16 skipping. Based on the results obtained using our minigenes, we employed CRISPR/Cas9 and introduced the identified human intronic regulatory sequence, including the BP, into the mouse genome. We generated two mutant mouse strains, one containing the human intronic regulatory sequence, termed "humanized strain," and one in which the mouse sequence had simply been deleted, termed "deletion strain" (Fig 6A). The intron–exon structure was retained for both strains, as only a part of the intron was exchanged or deleted, leaving all splice sites intact. Both strategies led to an altered *Camk2b* splicing pattern in the brain of the mutant mice, revealed in particular by the emergence of the primate-specific splice isoform Δ13,16 (Fig 6B). Sanger sequencing also confirmed the presence of the other previously identified primate-specific exon 16 exclusion isoforms. Interestingly, both mouse strains showed an additional band for a Δ13,16,17 isoform that we previously did not detect in human cells. These findings are in line with results from the minigene splicing assays and the postulated model of species-specific differences in BP strength. Furthermore, they corroborate that primate-specific *Camk2b* splicing is *cis*-regulated, with the mouse sequence harboring a functionally relevant, strong BP motif that leads to exon inclusion in the WT context. The sequence of the human intron contains a weak BP, and its knock-in into the mouse genome has a similar effect as simply deleting the strong mouse BP. In both cases, we observed a strong effect on splicing, especially in homozygous animals that lacked the FL and Δ17 isoforms. Instead, exon 16 was efficiently skipped in these animals, as revealed by the strong presence of the Δ16 and Δ16,17 isoforms (Fig 6B). These data

---

hom, homozygous animals. **(C)** Sashimi plot from STAR showing the alternative splicing of *Camk2b* exon 16 in RNA-Seq data from WT, heterozygous, and homozygous mice of the humanized strain. Each graph summarizes RNA-Seq data of four biological replicates. **(D)** Volcano plot mapping the differences in percent spliced in of cassette exons of homozygous versus WT animals against their respective *P*-values. Individual splicing events affecting *Camk2b* exon 16 are labeled. **(E)** Volcano plot mapping gene expression changes in the mouse model for both heterozygous and homozygous animals of the humanized strain against their respective *P*-values. **(F)** Example traces showing average of baseline and potentiated field excitatory postsynaptic potentials 30 min after LTP induction. Scale bar: 0.2 mV/5 ms. **(G)** Time course of LTP induction in CA3-CA1 synapses in acute hippocampal slices. LTP was induced after 10 min with a single train of 100 Hz, 1 s. WT: 15 slices, 6 mice; humanized (humanized strain, homozygote): 12 slices, 6 mice; and deletion (deletion strain, homozygote): 15 slices, 6 mice. **(H)** Dot plots depicting the field EPSP slope 30 min after LTP induction. **P < 0.01 from ANOVA followed by Dunnett's post hoc test. Also see Fig S7.

are consistent with the minigene results, but also point to additional regulatory layers in the in vivo situation, as the impact on exon skipping is stronger than in the minigene context. To confirm these observations, we performed RNA-Seq on cerebellum samples from the humanized strain. This analysis showed almost 100% inclusion of exon 16 in the WT mice, which was reduced to around 50% in heterozygous animals and essentially absent in homozygous animals (Fig 6C). We also checked whether alternative splicing was affected on a global level in the humanized mouse strain of our mouse model. However, only exon 16 of *Camk2b* was found to be substantially and significantly differentially spliced (Fig 6D), suggesting that alternative splicing is not globally affected in the humanized mouse strain. We also did not detect any significant differences in gene expression levels between WT and heterozygous animals, but detect only minor differences between WT and homozygous animals (Fig 6E). These results suggest that, under resting conditions, neither global gene expression nor global alternative splicing is significantly altered in our humanized *Camk2b* mouse model.

### Mice with altered *Camk2b* splicing pattern show reduced LTP

Having established mouse models for *Camk2b* exon 16 skipping, we next set out to determine whether these changes had an effect on synaptic plasticity. We performed an electrophysiological characterization of CA3-CA1 synapses in acute hippocampal slices of homozygous humanized or BP-deleted strains. Basal synaptic transmission, and short-term plasticity, measured as paired-pulse ratio (PPR), was unaltered in the mutant mice (Fig S7A and B, mean ± SD wt: 1.33 ± 0.15, humanized strain 1.29 ± 0.21; deletion strain 1.27 ± 0.09). However, high frequency–induced LTP was significantly impaired in both mouse strains 30 min poststimulation (Fig 6F–H, normalized amplitude wt: 1.285 ± 0.20, humanized strain: 1.09 ± 0.14; deletion strain: 1.07 ± 0.12). Induction of LTP with a single high-frequency tetanic pulse or with multiple pulses led to similar results (Fig S7C and D). In contrast, short-term potentiation, measured as the immediate response after the tetanic pulse (post-tetanic potentiation), was not affected (mean ± SD wt: 2.11 ± 0.53, humanized strain 2.10 ± 0.29; deletion strain 2.18 ± 0.41). Together, these observations show that in our mouse model with altered *Camk2b* alternative splicing, neither basal synaptic transmission nor short-term plasticity is affected, whereas LTP is severely impaired. As we did not alter coding sequences but only replaced an intronic splicing-regulatory element, our data provide evidence for a prominent role of *Camk2b* alternative splicing, which is controlled in a species-specific manner, in controlling synaptic plasticity, the molecular basis for elaborate cognitive functions. Whether this phenotype is caused by an altered *Camk2b* isoform ratio, the presence of the primate-specific isoforms, or the lack of the full-length isoform in our mouse model remains to be investigated in future research.

## Discussion

Species-specific alternative splicing has been suggested to contribute to shaping species-specific characteristics and abilities,

including cognitive abilities. However, how species-specific alternative splicing patterns are established is not well understood. In addition, how these patterns translate into species-specific functionality at the level of protein isoforms, cells, and whole organisms is another fundamental, largely unanswered question. Here, we uncover a pervasive mechanism underlying species-specific alternative splicing: the species-specific degree of deviation of BP sequences from consensus motifs. We also demonstrate that species-specific *CAMK2B* alternative splicing is controlled by fine-tuning a BP sequence and that *CAMK2B* alternative splicing correlates with crucial changes in neuronal functions linked to learning and memory.

Previously, we demonstrated how the strain-specific splicing of the *Camk2.1* gene in a marine insect controls the circadian timing of the species behavior (Kaiser et al, 2016). Mammalian *CAMK2B* is predominantly involved in the regulation of synaptic plasticity, and previous studies hinted at functional relevance of its alternative splicing (Brocke et al, 1995; GuptaRoy et al, 2000; Bayer et al, 2002; O'Leary et al, 2006; Bhattacharyya et al, 2020; Sloutsky & Stratton, 2021). Our results show how the primate-specific weakening of a BP motif in the *CAMK2B* gene leads to primate-specific exon 16 skipping and the generation of several primate-specific protein isoforms. In line with previous studies (Barbosa-Morais et al, 2012; Gao et al, 2015), changes in a *cis*-acting element, rather than the *trans*-acting environment, control the observed species-specific splicing differences. Interestingly, rather than affecting auxiliary enhancer or repressor sequences, the identified genomic differences affect the BP sequence, one of the canonical splicing motifs. Thus, alteration of BP strength can contribute to the decoupling of alternative splicing from changes in the *trans*-acting environment. As the *trans*-acting environment differs between different organs and tissues, our findings provide a further explanation for species-specific splicing patterns that are present throughout different organs.

The BP is a prime target for what has been termed "evolutionary tinkering" (Jacob, 1977; Ule & Blencowe, 2019), meaning the gradual accumulation of mutations that promote new protein functions with minimal disruptive effects on existing ones. Introns often contain multiple functional BPs, leading to flexibility regarding BP choice, which may facilitate the evolutionary adaptation of individual BPs. In addition to this, introns can be removed in multiple steps, a process termed recursive splicing (Wan et al, 2021). This process may facilitate increased variation of individual BP sequences that alter splicing efficiency without disrupting splicing altogether.

We also show that primate-specific CaMKIIβ protein isoforms differ subtly in their kinetic properties and in their substrate spectra. Kinetic differences are presumably mediated by conformational differences between various inactive and active states of the holoenzyme. However, the exact nature of these states is currently still debated (Chao et al, 2011; Myers et al, 2017; Sloutsky et al, 2020; Buonarati et al, 2021). In our study, we confirm recent results showing that, under steady-state conditions, the alternative splicing-impacted variable linker segment does not affect the cooperativity of the enzyme (Sloutsky et al, 2020) but instead modulates the maximal activity of CaMKIIβ at optimal calmodulin concentrations. Although the observed effects are small, CaMKII

isoforms represent the most abundant proteins at the postsynapse (Erondu & Kennedy, 1985; Cheng et al, 2006), which means that even small kinetic differences may translate into a large overall effect in vivo.

Similar to a previous publication (Bhattacharyya et al, 2020), we find isoform-specific differences in CaMKII autophosphorylation in our analog-sensitive kinase assay. Exon 13 exclusion isoforms exhibit a down-regulation of the inhibitory autophosphorylations (T306/307), which prevent reassociation of calmodulin and thereby the full activation of the enzyme. The complementary activating autophosphorylation (T287) occurs in all isoforms, but to a smaller extent in exon 16 exclusion isoforms. These findings corroborate and expand on a study of fly CaMKII that revealed isoform-specific differences in in vitro substrate specificity with isolated proteins (GuptaRoy et al, 2000), suggesting direct interactions of the linker segment with selected target proteins.

In addition, our MS data suggest the presence of many novel CaMKIIβ substrates (Supplemental Data 2), often featuring tyrosine phosphorylations, which had previously only been reported for an artificial CaMKII construct (Sugiyama et al, 2008). Although isoform- and group-exclusive phosphorylation targets exist, the isoforms also target many overlapping sites. An additional difference between the substrate spectra lies in the relative abundance of the various phosphorylation sites, showing that a given substrate has a different probability of being phosphorylated by the different CaMKIIβ isoforms. Being identical among CaMKIIβ variants, the kinase domains per se cannot be the source of these differences. However, the flexible linker, modulated by alternative splicing, can change the probability that a particular substrate comes in contact with the active center.

CaMKIIβ also plays a structural role in synapses. Alternative splicing changes the affinity of the resulting isoforms to actin (O'Leary et al, 2006). Because of the high abundance of CaMKIIβ in neurons, this likely alters the overall architecture of the cytoskeleton (Hoffman et al, 2013) and presumably of other protein structures, such as the postsynaptic density. It is therefore likely that the length and composition of the variable linker affect the positioning of CaMKIIβ isoforms within these structures and hence the exposure to specific substrates. Although CaMKIIβ readily dissociates from actin filaments after stimulation (Shen & Meyer, 1999; Lin & Redmond, 2008), it has been proposed that because of the transient nature of neuronal signaling, every CaMKII subunit only phosphorylates a single substrate during an individual calcium spike (Bayer & Schulman, 2019), emphasizing the impact of initial differences in subcellular localization.

All of the points discussed above emphasize that primate-specific CaMKIIβ enzyme isoforms, with their specific functionality, may play a unique role in the primate brain. In an attempt to address the in vivo functionality, we generated a mouse model for *CAMK2B* alternative splicing. In these mouse mutants, the normal balance of splice isoforms is disrupted leading to a strong defect in LTP. Both of our mouse model strains, although having a slightly different genotype, show an identical phenotype with respect to both transcriptome changes and LTP characteristics. This observation underscores a direct causal link between differences in *CAMK2B* alternative splicing and functional consequences for synaptic plasticity. However, further work is required to investigate

whether this phenotype is due to an imbalance of splicing isoforms in general, the presence of the primate-specific isoforms, or rather the (almost) complete loss of the FL isoform. Although a complete *Camk2b* knockout in mice leads to loss of LTP under some conditions (Borgesius et al, 2011), it seems unlikely that all other splice isoforms that are present in our mouse mutant are not able to compensate for the loss of the full-length variant. This suggests an important role of the ratio of *CAMK2B* splice variants and/or the presence of particular isoforms in controlling LTP.

Our work provides an example of a mouse model in which only a species-specific splice-regulatory *cis*-acting element has been mutated, an approach which holds great promise in deciphering the exact mechanistic framework of splicing regulation and its functional consequences. The observed effect on exon 16 splicing can potentially also be induced by other mutations, including single-nucleotide polymorphisms, in the splice sites. We thus expect that single-nucleotide polymorphisms or other mutations that modulate *CAMK2B*, and potentially *CAMK2A*, alternative splicing alter *CAMK2* functionality and may be involved in a variety of neurological diseases. Taken together, we connect weakening of the BP sequence with species-specific alternative splicing and present a mouse model that connects *CAMK2B* alternative splicing with LTP, with implications for the generation of species-specific cognitive abilities.

# Materials and Methods

### Identification of endogenous *Camk2* alternative splicing isoforms

RT–PCR was performed with total RNA from human cerebellum (Cat# 636535; Clontech), and mouse cerebellum, frog brain tissue, and rhesus macaque total brain RNA (Cat# UR-201; Zyagen). Human cerebellum RNA contained material pooled from three male Asians, aged 21–29 (information provided by supplier). Total cerebellum RNA from mouse (*M. musculus*) and total brain tissue RNA from frog (*X. laevis*) were extracted via TRIzol (see below). Where necessary, specificity for *CAMK2B* was inferred by a gene-specific RT primer, annealing to the less conserved exon 25 (numbering based on scheme in Fig 1A, human: TTG TGG TTG TCG TCG TCA TC; mouse: ACG AGG CAG ACA CAA ACA TG). Primers for the splice-sensitive radioactive PCR annealed to exons 9 and 23 (human/macaque for: CTC CAC GGT AGC ATC CAT GA; rev: AGT CCA TCC CTT CAA CCA GG; mouse for: CCA CCG TGG CCT CTA TGA T; rev: AAT CCA TCC CTT CGA CCA GG; and Xenopus for: CCA CTG TTG CTT CCA TGA TG; rev: CCT GGT AGA AGG GAT AGA CT). PCR products were sequenced using the CloneJET PCR Cloning Kit (Thermo Fisher Scientific). RT–PCRs with radioactively labeled forward primers and quantification of PCR products were performed as previously described (Haltenhof et al, 2020).

### Minigene design and splicing assays

Minigenes were designed using the pcDNA3.1+ vector backbone. The minigenes contained the following sequences: exon 11 with 300 bp of the downstream intron, exon 16 with 100 bp of the upstream intron, the full intron in between exons 16 and 17, exon 17 with

300 bp of the downstream intron, and exon 23 with 100 bp of the upstream exon (see Fig S1B). Alternative splicing of the minigenes was analyzed in N2A, SH-SY5Y, HEK, and HeLa cells in biological triplicates. HEK and HeLa cells were cultivated in DMEM high-glucose medium (Biowest) with 10% FBS and penicillin–streptomycin (Biowest). N2A cells were cultivated in a 1:1 mix of Opti-MEM and DMEM (Opti-MEM with GlutaMAX; Gibco and DMEM with GlutaMAX; Gibco). SH-SY5Y cells were cultivated in DMEM high-glucose medium with 10% FBS, penicillin–streptomycin, and additional L-glutamine (1% vol/vol of 200 mM). Cells were seeded in 12-well plates with a concentration of $1 \times 10^5$ cells/well (HEK, SH-SY5Y, N2A) or $1.5 \times 10^5$ cells/well (HeLa). After 24 h, the cells were transfected with 1 μg plasmid and 2 μl Roti-Fect (Carl Roth GmbH) transfection reagent per well. Cells were harvested 48 h after transfection, and RNA was extracted using RNA Tri-Liquid (BioSell) reagent according to the manufacturer's instruction. DNase I (Epicentre) digestion was performed according to the manufacturer's instruction to minimize contamination with plasmid DNA. Alternative splicing was analyzed by radioactive RT–PCR as described above, with a vector-specific primer pair (T7f: TAA-TACGACTCACTATAGGG, BGHr: CCTCGACTGTGCCTTCTA).

### Prediction of BP sequences

The SVM-BPfinder (Corvelo et al, 2010) tool was used to predict BP sequences. If not otherwise specified, human was selected as target organism to predict BP strength.

### Expression and purification of selected CaMKIIβ isoforms

Selected CaMKIIβ isoforms were expressed in High Five insect cells via the baculovirus system. All purification steps were performed at 4°C. Cell pellets were resuspended in CaMKII lysis buffer (10 mM Tris–HCl, pH 7.5, 500 mM NaCl, 1 mM EDTA, 1 mM EGTA, 5% glycerol, and 1 mM DTT) supplemented with protease inhibitors (cOmplete; Roche) and lysed by sonication. Insoluble particles were separated by centrifugation at 21,500 rpm for 1 h (JA-25.50 fixed-angle rotor; Beckman Coulter). The soluble fraction was incubated with Strep-Tactin Sepharose beads (IBA Lifesciences) for 1 h and washed with CaMKII lysis buffer. Bound protein was eluted with CaMKII SEC buffer (50 mM Pipes, pH 7.5, 500 mM NaCl, 1 mM EGTA, 10% glycerol, and 1 mM DTT) containing 2.5 mM desthiobiotin (IBA Lifesciences). Eluted protein was concentrated and run on a Superose 6 10/300 Gl size-exclusion column (Cytiva) with CaMKII SEC buffer. Fractions were pooled according to SDS–PAGE and chromatogram, concentrated to ~1 mg/ml, and flash-frozen in single-use aliquots in liquid nitrogen. Before use, aliquots were thawed on ice, gently mixed by pipetting, and centrifuged at 20,000 rcf for 5 min. Exactly equal concentrations were determined by repeated SDS–PAGE, Coomassie staining, and quantification with ImageQuant TL (Cytiva).

### Expression and purification of CaMKII substrate Syntide 2–GST

The sequence for Syntide 2 (PLARTLSVAGLPGKK) was expressed as a GST fusion protein, with a TEV-cleavable N-terminal His-tag. A short linker (GGGGSGGGGS) was inserted between the Syntide 2 sequence and the C-terminal GST-tag. The fusion protein was expressed in

BL21 RIL cells using autoinduction medium. All purification steps were performed at 4°C. Cell pellets were resuspended in lysis buffer (50 mM Tris–HCl, pH 7.5, 150 mM NaCl, 20 mM imidazole, and 1 mM DTT) containing protease inhibitors (cOmplete; Roche) and lysed by sonication. Insoluble particles were separated by centrifugation at 21,500 rpm for 1 h (JA-25.50 fixed-angle rotor; Beckman Coulter). The soluble fraction was loaded on a HisTrap Crude column (Cytiva) and eluted with a linear gradient of elution buffer (20 mM Tris–HCl, pH 7.5, 300 mM NaCl, 500 mM imidazole, and 1 mM DTT). Target fractions were pooled, supplied with TEV protease (self-made), and dialyzed against lysis buffer overnight. Digested samples were rerun on a HisTrap Crude column. The flow-through was collected, concentrated, and run on a HiLoad Superdex 75 26/60 size-exclusion column (Cytiva), equilibrated with SEC buffer (20 mM Pipes, pH 7.5, and 50 mM NaCl). Target fractions were pooled, concentrated to 22 mg/ml, and flash-frozen in liquid nitrogen.

### Expression and purification of human FL τ (τ-441)

Human FL τ (τ-441) was expressed as a fusion protein with an N-terminal His- and a C-terminal StrepII-tag. The protein was expressed in BL21 RIL cells in TB medium. Bacteria were grown at 37°C until an optical density of 0.6–0.8. Protein expression was induced with 1 mM IPTG for 3 h at 37°C. Cell pellets were resuspended in PBS buffer supplemented with 5 mM imidazole and protease inhibitors (cOmplete; Roche). Cells were lysed by sonication and incubated at 80°C in a water bath for 10 min with sporadic manual agitation. The lysate was cooled on ice for 10 min and supplemented with fresh protease inhibitors and 2 mM DTT. The lysate was cleared by centrifugation at 21,500 rpm for 30 min (JA-25.50 fixed-angle rotor; Beckman Coulter). The supernatant was loaded onto a HisTrap FF Crude 5 ml column (Cytiva) equilibrated with PBS supplemented with 5 mM imidazole and 1 mM DTT. The column was washed until baseline and the protein eluted with a linear gradient from 5 to 500 mM imidazole. Fractions were pooled based on the chromatogram and SDS–PAGE. Pooled fractions were loaded on a StrepTrap 5 ml column (Cytiva), equilibrated with PBS + 1 mM DTT. The column was washed until baseline, and the protein was eluted with PBS containing 1 mM DTT and 2.5 mM desthiobiotin (IBA Lifesciences). Fractions were pooled based on the chromatogram and SDS–PAGE. The pooled sample was concentrated using a molecular weight cutoff of 3 kD and run on a Superdex S200 26/60 (GE), equilibrated in PBS supplemented with 1 mM DTT. Fractions were pooled based on the chromatogram and SDS–PAGE, concentrated to ~15 mg/ml, and flash-frozen in liquid nitrogen.

### In vitro kinase assay

The protocol was adapted from Coultrap & Bayer (2012). CaMKII activity was measured by $^{32}P$ incorporation into the substrate Syntide 2–GST or τ-441 (human). The model substrate Syntide 2 (Hashimoto & Soderling, 1987) was linked to GST to increase its molecular weight, facilitate purification, and enable separation on an SDS–PAGE. Reactions were performed in 0.2 ml PCR stripes. Purified CaMKIIβ was diluted to 10 nM in a mix containing 50 mM Pipes, pH 7.2, 0.1% BSA, 2 mM CaCl$_2$, 10 mM MgCl$_2$, 50 μM Syntide 2–GST, or 10 μM τ (human τ-441). The reaction was started by adding

1 nM to 4 $\mu$M calmodulin (Calbiochem) and 100 $\mu$M ATP (~1 Ci mmol$^{-1}$ [$\gamma^{32}$P]-ATP). Reagents were pre-incubated at 30°C for 5 min. Reactions were carried out in a final volume of 30 $\mu$l for 2 min at 30°C. Reactions were terminated by adding 10 $\mu$l SDS sample buffer. Samples were run on a 12.5% SDS–PAGE, dried, and analyzed via a photostimulable phosphor plate. Gels were quantified using ImageQuant 5.2 or ImageQuant TL (Cytiva). Results were plotted using GraphPad Prism 6 and fit to a Hill equation (allosteric sigmoidal non-linear fit). For the standard in vitro kinase assay, the experiment was repeated two times in triplicates. To compare the maximal activity at optimal calmodulin concentrations, V/Vmax$_{FL}$ values for calmodulin concentrations from 100 to 1,000 nM were pooled and plotted using GraphPad Prism 6. Normal distribution and equality of variances were tested via the Shapiro–Wilk test, Q-Q plots, and the F test. Based on the results, a $t$ test or Welch's $t$ test was performed. Resulting $P$-values were adjusted for multiple comparisons using Holm's method. Statistical analysis was performed in R and RStudio. For the autoactivity assay, the activation of CaMKII with varying concentrations of calmodulin was performed in the absence of the substrate protein. After a 2-min incubation, the activation was quenched by addition of 5.3 mM EGTA. The substrate protein was added together with 3.3 mM MgCl$_2$ to enable the phosphorylation reaction. The sample was incubated for 3 min and the reaction terminated with SDS sample buffer. The analysis was performed as described above.

### Analog-sensitive kinase, pulldown, and in vitro kinase assays

HEK cells were cultured as described above and seeded at a concentration of 0.2 × 10$^5$ cells/ml and 12 ml in 10-cm dishes or 30 ml in 15-cm dishes. The CaMKII$\beta\Delta$13,16 analog-sensitive variant was transfected 24 h after seeding as described above, using 12 $\mu$g for 10-cm dishes and 36 $\mu$g for 15-cm dishes. 24 h after transfection, cells were harvested with trypsin, transferred to 1.5-ml reaction tubes, and washed with PBS before being flash-frozen in liquid nitrogen and stored at −80°C. Cell pellets corresponding to 3 × 10 cm dishes and 2 × 15 cm dishes were thawed on ice and resuspended in CaMKII lysis buffer (10 mM Tris–HCl, pH 7.5, 500 mM NaCl, 1 mM EDTA, 1 mM EGTA, 5% glycerol, and 1 mM DTT) supplemented with protease inhibitors (cOmplete EDTA-free; Roche). Cells were lysed by sonication on ice at 40% amplitude, 0.5 cycle, and six rounds of 5 s. Lysates were cleared by centrifugation at 20,000 rcf for 30 min at 4°C. The supernatant was transferred to a new reaction tube and mixed with 50 $\mu$l pre-equilibrated Strep-Tactin XT beads (IBA) and supplemented with biotin-blocking solution (IBA). Samples were incubated for 1 h at 4°C with slow rotation. Beads were sedimented by centrifugation at 500 rcf for 5 min at 4°C. Beads were washed three times in CaMKII SEC buffer (50 mM Pipes, pH 7.5, 500 mM NaCl, 1 mM EGTA, 10% glycerol, and 1 mM DTT), and bound protein eluted CaMKII SEC buffer supplemented with 50 mM biotin (IBA). The eluate was dispersed into single-use aliquots, flash-frozen in liquid nitrogen, and stored at −80°C. To compare the AS variant with the wt kinase, a standard in vitro kinase assay was performed as described above, using a limited range of calmodulin concentrations and roughly estimating the concentration via UV absorption at 280 nm. To test the inhibition by various ATP analogs, the standard IVK assay was modified and set to a single calmodulin concentration of

100 nM. The reaction mixture contained varying concentrations of non-radioactive ATP (0–1 mM) or 0.5 mM of one of the following non-radioactive ATP analogs: N$^6$-methyl-ATP, N$^6$-etheno-ATP, N$^6$-phenyl-ATP, and N$^6$-benzyl-ATP (Jena Bioscience).

### Analog-sensitive kinase assay—in vivo labeling and thiophosphate enrichment

The thiophosphate enrichment strategy was based on Michowski et al (2020), with modifications. The analog-sensitive kinase variants were PCR-amplified with primers omitting the Twin-Strep-tag and cloned back into the pcDNA3.1 expression plasmid, to avoid interference from the affinity tag. N2A cells were cultured as described above and seeded into 15-cm dishes at a concentration of 0.1 × 10$^6$ cells/ml and 30 ml/dish. Cells were incubated for 24 h and transfected with 37.5 $\mu$g DNA and 75 $\mu$l Roti-Fect (Carl Roth GmbH) per 15-cm dish, as described above. Cells were grown for 48 h, and washed with 20 ml PBS and subsequently 20 ml AS lysis buffer (20 mM Pipes, pH 7.5, 150 mM NaCl, 10 mM MgCl$_2$, and 1 mM EGTA). The liquid was removed, and the dish was carefully washed with 1.2 ml AS lysis buffer, supplemented with protease inhibitors (cOmplete; Roche), phosphatase inhibitors (PhosSTOP; Roche), and 0.5 mM TCEP. The liquid was removed thoroughly, and the cells were detached with a cell scraper, transferred to a reaction tube, and kept on ice until all samples had been harvested. From then on, samples were processed in parallel in Protein LoBind tubes (Thermo Fisher Scientific). Each 15-cm dish resulted in ~1.2 ml cell suspension, which was split into two 600 $\mu$l aliquots. The remaining cells were discarded. Each aliquot was supplemented with 75 $\mu$l detergent mix (3.6% nOG and 36 mM CaCl$_2$) and briefly mixed. The labeling reaction was started by addition of 225 $\mu$l reaction mix (200 nM calmodulin, 0.1 mM N$^6$-benzyl-ATP$\gamma$S, 0.2 mM ATP, 3 mM GTP, and PhosSTOP phosphatase inhibitors in AS lysis buffer) and incubated for 30 min at 30°C with sporadic manual agitation. For the untransfected control, calmodulin was omitted. The reaction was terminated by addition of EDTA/EGTA to a final concentration of 10 mM each. Labeled samples were briefly sonicated to create a homogeneous suspension, and concentrations were determined by Pierce 660 nM assay (Thermo Fisher Scientific). Samples were flash-frozen in liquid nitrogen and stored at −80°C. For the Western blot, aliquots were alkylated with 50 mM PNBM (p-nitrobenzyl mesylate; Agilent Technologies) at a final concentration of 2.5 mM for 1 h at RT. The reaction was terminated by addition of SDS sample buffer (containing DTT) and the samples analyzed via standard SDS–PAGE and semi-dry Western blotting. The blot was developed using an anti-thiophosphate ester antibody (ab92570; Abcam) and an HRP-linked anti-rabbit antibody (Cell Signaling Technologies).

For thiophosphate enrichment, samples were thawed and lysate corresponding to 6 mg protein was transferred into a 15-ml tube for protein precipitation. All samples were equalized in volume with AS lysis buffer and supplemented with 5 volumes of ice-cold methanol/chloroform mix (ratio 4:1), followed by 3 vol of ice-cold H$_2$O. The samples were thoroughly mixed, incubated for 10 min on ice, and centrifuged for 20 min at 2,000 rcf. The resulting pellet, located at the interface, was washed in 5 vol of ice-cold methanol and centrifuged for 20 min at 2,000 rcf. The supernatant was removed and the pellet dried at RT. The dried pellet was

resuspended in 800 $\mu$l denaturation buffer (100 mM NH$_4$HCO$_3$, 2 mM EDTA, 10 mM TCEP adjusted to pH 7–8, and 8 M urea), adjusted to 6 M urea with H$_2$O, and incubated at 55°C for 1 h with agitation at 300 rpm (JA-25.50 fixed-angle rotor; Beckman Coulter). The sample was slowly cooled to RT for 10 min and diluted to 2 M urea with 50 mM NH$_4$HCO$_3$ in H$_2$O. TCEP (pH adjusted to 7–8) was added to a final concentration of 10 mM. Trypsin (Trypsin, TPCK treated, from bovine pancreas; Sigma-Aldrich) was added at a ratio of 1:20 (w/w, based on starting material), and the samples were digested overnight at 37°C. The next morning, 10 M NaOH was added to a final concentration of 0.08 mM and the digestion continued for 3 h. The digest was acidified with 2.5% TFA to a final concentration of 0.1% and a pH of ~2.5. If required, more TFA was added to lower the pH. The digest was centrifuged for 3 min at 1,400 rcf and the supernatant aliquoted to a new tube. Sep-Pak Plus cartridges (Waters) were equilibrated by sequential washing with 10 ml 0.1% TFA/50% acetonitrile (in H$_2$O) and 10 ml 0.1% TFA (in H$_2$O). The sample was loaded by passing it through the cartridge five times. The cartridge was washed with 10 ml 0.1% TFA (in H$_2$O). Bound peptides were eluted with 4 ml 80% acetonitrile/0.1% acetic acid and dried overnight in a vacuum centrifuge. SulfoLink beads (Thermo Fisher Scientific) were transferred to a Protein LoBind tube and washed with 200 mM Hepes, pH 7.0. Beads were incubated with 200 mM Hepes, pH 7.0, and 25 $\mu$g/ml BSA for 10 min at RT in the dark. Beads were sequentially washed with 200 mM Hepes, pH 7.0, and two times with 4 M urea, 0.1 M Tris, pH 8.8, and 10 mM TCEP (pH of stock solution ~2.5; this lowers the total pH to ~8.0). The dried peptides were resuspended in 4 M urea, 0.1 M Tris, pH 8.8, and 10 mM TCEP and acidified to pH 5 with 5% (vol/vol) formic acid. The peptide solution was added to the equilibrated beads and rotated overnight at RT in the dark. The next day, the beads were centrifuged at 2,000 rcf for 3 min, and the supernatant was discarded. The beads were washed sequentially with 4 M urea in 20 mM Hepes, pH 7.0, H$_2$O, 5 M NaCl, 50% acetonitrile in H$_2$O, and 5% (vol/vol) formic acid. Unused binding sites were blocked by incubation with a fresh solution of 10 mM DTT for 10 min in the dark. Bound peptides were eluted in three steps with a solution of 2 mg/ml Oxone (potassium peroxymonosulfate; Sigma-Aldrich) in H$_2$O. Eluates were pooled and desalted using Sep-Pak Plus cartridges as described above. Samples were dried in a vacuum centrifuge and stored at −80°C.

To remove remaining contaminants, peptides were further purified with styrene–divinylbenzene StageTips. StageTips were prepared by inserting the material into standard 200 $\mu$l pipette tips and washing sequentially with methanol, and 80% acetonitrile in 0.1% formic acid, and in two steps with 0.1% formic acid in H$_2$O. The resuspended samples were acidified with 10% formic acid to a final concentration of 1%. The samples were loaded and passed through the StageTips, followed by sequential washing with 0.1% formic acid in H$_2$O and two rounds of 80% acetonitrile in 0.1% formic acid. Bound peptides were eluted with 5% NH$_4$OH in 60% acetonitrile, split into two equal aliquots, and dried in a vacuum centrifuge. Peptides were measured on an Orbitrap Q Exactive HF (Thermo Fisher Scientific) or an Orbitrap Exploris 480 (Thermo Fisher Scientific). MS raw data were analyzed using MaxQuant (version 1.6.5.0) against the UniProt mouse reference proteome (downloaded in November 2021, mouse, 25,367 entries). Subsequent analysis was

done in Python (version 3.8.5; Anaconda Distribution) using the packages pandas, NumPy, matplotlib, seaborn, upsetplot, SciPy, and sklearn. Contaminants and reverse peptide hits were removed, and the analysis was restricted to phosphorylated peptides with a localization probability ≥ 0.75. The overlap between the two datasets was calculated using the unique phosphosite (protein/ gene name + identity of phosphorylated residue) as an index. The intensity values of both datasets were normalized before merging, using the min–max normalization: $x_{norm.} = \frac{x - min(x)}{max(x) - min(x)}$. Min(x) and max(x) were set to the respective minimal and maximal value of the individual datasets. When pooling replicates, an average intensity value was calculated. If only one replicate featured an intensity value for the respective target, this value was kept. Correlation matrices were calculated using a Pearson correlation coefficient.

The MS data have been deposited to the ProteomeXchange (Perez-Riverol et al, 2021) Consortium via the PRIDE partner repository with the dataset identifier PXD035346.

## Generation of the mouse models

Mouse models were generated in the Transgenics Facility at the Max Delbrück Center for Molecular Medicine Berlin (MDC) under the supervision of Dr. Ralf Kühn. The models were based on the *CAMK2B* minigenes. CRISPR/Cas9 was used in C57BL/6 mouse zygotes as described (Wefers et al, 2017) to remove the 100 bp initially found to harbor the *cis*-acting element in the endogenous mouse *Camk2β* gene. A synthetic gene was used as a repair template to insert the human ortholog of the excised sequence into the endogenous mouse gene (humanized strain). A deletion strain was generated in which the repair process failed and only the mouse sequence was deleted. Mice were handled according to institutional guidelines under experimentation licenses G0111/17-E65, T0100/03, and T0126/18 approved by the Landesamt für Gesundheit und Soziales and housed in standard cages in a specific pathogen-free facility on a 12-h light–dark cycle with ad libitum access to food and water.

## RNA-seq analysis

### Mouse model

Total RNA was extracted from mouse cerebellum tissue as described above (minigene splicing assay). Four male wt, two male and two female heterozygous, and four male homozygous mice of the humanized strain were selected for RNA-Seq. For library preparation, DNase I–digested RNA samples were filtered using the polyA+ selection method at BGI Genomics and sequenced using DNBSeq PE150 sequencing. This yielded ~50–60 million paired-end 150-nt reads. Reads were aligned to the GRCm38 genome using the STAR aligner (v.2.7.9a) (Dobin et al, 2013), yielding on average ~75% uniquely mapped reads. Files were indexed using SAMtools (Danecek et al, 2021), and the splicing pattern was analyzed using rMATS (v3.1.0) (Shen et al, 2014). Downstream analyses and data visualization were performed using standard python code (v3.8.5). Data were visualized and sashimi plots generated via IGV (Robinson et al, 2011). Gene expression patterns were analyzed using Salmon (v1.8.0) (Patro et al, 2017) and DESeq2 (Love et al, 2014). Volcano plots

were generated using GraphPad Prism 5–6. RNA-sequencing data generated in this study are available under GEO #GSE208181.

### Other mammals

Publicly available RNA-Seq data were analyzed for various mammals. For human, chimpanzee (*Pan troglodytes*), bonobo (*Pan paniscus*), and rhesus macaque (*M. mulatta*), data from cerebellum white tissue and cerebellum gray tissue from multiple individuals were selected. For gibbon (*Hylobates lar*), data from different brain regions from a single individual were selected. For gorilla (*Gorilla gorilla*) and orangutan (*Pongo pygmaeus*), data from cerebellum and total brain tissue were selected. For pig (*Sus scrofa*), data from cerebellum tissue were selected. Reads were aligned to the respective genome (human: GRCh38; chimpanzee: panTro6; bonobo: panPan1.1; gorilla: gorGor6; orangutan: ponAbe3 [*Pongo abelii*]; Gibbon: nomLeu3 [*Nomascus leucogenys*]; rhesus macaque: rheMac10; and pig: SusScr11) using the STAR aligner (v.2.7.9a) (Dobin et al, 2013). Subsequent analysis was performed as described above. To calculate % skipped values for *CAMK2B* exon 16, the sum of all individual exon 16 skipping events was calculated. For final visualization, cerebellum gray and white matter files (where available) were merged to create combined cerebellum files. A list of all used publicly available RNA-Seq data, including species, tissue, read length, and used reference genome, can be found in Table S2.

### Identification and analysis of orthogonal exons

Orthogonal exons in human and mouse were identified using the liftOver tool from the UCSC Genome Browser (Kent et al, 2002), with custom optimization using the human genome assembly hg38 and the mouse genome assembly mm10. Publicly available RNA-Seq data from human brain tissue (47 samples from 35 individuals) and mouse brain tissue (nine samples from nine individuals) were analyzed as described above and restricted to cassette exon events. Only splicing events supported by at least three datasets were kept. The results were filtered for a SD of PSI below 0.2 and a minimal mean junction read count of 10. Alternative exons were defined as exons showing a PSI < 0.9, and constitutive exons, as exons showing a PSI > 0.9. If orthologous alternative exons were identified in multiple transcripts, with different upstream or downstream exons, only the first listed entry was kept. Species-exclusive exons were defined as those being alternative in one, and constitutively included in the other species. If indicated, a further threshold of a minimal difference in PSI levels of 0.2 was applied. BP scores were calculated using SVM-BP (Corvelo et al, 2010), and splice site scores, using MaxEntScan (Yeo & Burge, 2004). The difference between means was calculated using the paired Wilcoxon signed-rank test.

### Electrophysical characterization

All experiments regarding the electrophysical characterization were entirely performed in the research group of Prof. Dietmar Schmitz (Charité; NeuroCure) under the supervision of Dr. Alexander Stumpf. Hippocampal slices were prepared from adult C57BL/6J and transgenic (deletion and humanized) mice. Animals were anesthetized with isoflurane and decapitated. The brain was quickly removed and chilled in ice-cold sucrose-based artificial cerebrospinal fluid containing (in mM) the following: NaCl 87, NaHCO$_3$ 26, glucose 10, sucrose 50, KCl 2.5, NaH$_2$PO$_4$ 1.25, CaCl$_2$ 0.5, and MgCl$_2$ 3, saturated with 95% (vol/vol) O$_2$/5% (vol/vol) CO$_2$, pH 7.4. Horizontal slices (300 $\mu$m) were cut and stored and submerged in sucrose-based artificial cerebrospinal fluid for 30 min at 35°C and subsequently stored in ACSF containing (in mM) the following: NaCl 119, NaHCO$_3$ 26, glucose 10, KCl 2.5, NaH$_2$PO$_4$ 1, CaCl$_2$ 2.5, and MgCl$_2$ 1.3 saturated with 95% (vol/vol) O$_2$/5% (vol/vol) CO$_2$, pH 7.4, at RT. Experiments were started 1–6 h after the preparation.

Recordings were performed in a submerged recording chamber (Warner instruments RC-27L), filled with ACSF with solution exchange speed set to 3–5 ml/min at RT (22–24°C). Stimulation electrodes were placed in the *striatum radiatum* of CA1 (near CA3) to stimulate Schaffer collaterals. Recording electrodes were placed in the *striatum radiatum* of the CA1 field. Stimulation was applied every 10 s. In order to analyze the input–output relationship, stimulation intensities were adjusted to different FV amplitudes (0.05 mV increments, 0.05–0.4 mV) and correlated with the corresponding field excitatory postsynaptic potential (fEPSP). PPRs were determined by dividing the amplitude of the second fEPSP (50-ms inter-stimulus interval) by the amplitude of the first (average of 10 repetitions). LTP: Basal stimulation was applied every 10 s to monitor stability of the responses at least for 10 min before LTP was induced by one single high-frequency stimulation train (100 pulses, 100 Hz). Magnitude of LTP was determined by normalizing the average of the initial fEPSP slopes 25–30 and 55–60 min after LTP induction to average baseline fEPSP slope. Data collection and quantification was performed blindly. One-way ANOVA and Dunnett's multiple comparison test were used to compare the mean LTP and PPR values of the transgenic animals (humanized and deletion) with the WT control. LTP induction by multiple high-frequency trains was only performed in the humanized strain and in WT animals; thus, an unpaired *t* test was performed to compare these groups. Normal distribution of the data was tested via the D'Agostino & Pearson omnibus normality test.

## Data Availability

Material generated in this study is available upon reasonable request by email to F Heyd. RNA-Seq data generated in this study are available under GEO #GSE208181.

## Supplementary Information

## Acknowledgements

The authors would like to thank Iva Lucic and Andrew Plested for discussions regarding CaMKII functionality, Kevan Shokat for insights into the analog-sensitive kinase system, and members of the F Heyd and MC Wahl laboratories for discussing all aspects of the project. Matthis Jahnel performed initial bioinformatic analysis to identify orthologous exons. A Franz and AI

Weber were funded by PhD Fellowships of the Boehringer Ingelheim Fonds (BIF). Initial project funding was provided by FU research funding. This study was supported by a grant from the Deutsche Forschungsgemeinschaft (TRR186/A15; project number 278001972) to F Heyd and MC Wahl.

## Author Contributions

A Franz: formal analysis, investigation, visualization, methodology, and writing—original draft, review, and editing.

AI Weber: formal analysis, investigation, visualization, and writing—review and editing.

M Preußner: formal analysis and investigation.

N Dimos: investigation.

A Stumpf: formal analysis, investigation, and writing—original draft.

Y Ji: investigation.

L Moreno-Velasquez: investigation.

A Voigt: investigation.

F Schulz: investigation.

A Neumann: investigation.

B Kuropka: investigation.

R Kühn: supervision, investigation, methodology, and writing—review and editing.

H Urlaub: formal analysis, supervision, investigation, methodology, and writing—review and editing.

D Schmitz: formal analysis, supervision, investigation, visualization, methodology, and writing—review and editing.

MC Wahl: conceptualization, formal analysis, supervision, funding acquisition, investigation, methodology, and writing—original draft, review, and editing.

F Heyd: conceptualization, data curation, formal analysis, supervision, funding acquisition, investigation, visualization, methodology, and writing—original draft, review, and editing.

## Conflict of Interest Statement

The authors declare that they have no conflict of interest.

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
