## [Reviewer comments · Life Science Alliance]

Life Science Alliance

Branch point strength controls species-specific CamK2b alternative splicing and regulates LTP

Andreas Franz, A. Ioana Weber, Marco Preußner, Nicole Dimos, Alexander Stumpf, Yanlong Ji, Laura Moreno-Velasquez, Anne Voigt, Frederic Schulz, Alexander Neumann, Benno Kuroopka, Ralf Kühn, Henning Urlaub, Dietmar Schmitz, Markus Wahl, and Florian Heyd

DOI: <https://doi.org/10.26508/lsa.202201826>

Corresponding author(s): Florian Heyd, Freie Universität Berlin

Review Timeline:

Submission Date:	2022-11-12
Editorial Decision:	2022-11-14
Revision Received:	2022-11-28
Editorial Decision:	2022-11-29
Revision Received:	2022-12-01
Accepted:	2022-12-02

Transaction Report:

Please note that the manuscript was previously reviewed at another journal and the reports were taken into account in the decision-making process at Life Science Alliance. Since the original reviews are not subject to Life Science Alliance's transparent review process policy, the reports and author response cannot be published.

November 14, 2022

Re: Life Science Alliance manuscript #LSA-2022-01826-T

Florian Heyd
FU Berlin
BCP
Takustr. 6
Berlin 14195
Germany

Dear Dr. Heyd,

Thank you for submitting your manuscript entitled "Branch point evolution controls species-specific alternative splicing and regulates long term potentiation" to Life Science Alliance. We invite you to submit a revised manuscript addressing the Reviewer comments.

Thank you for this interesting contribution to Life Science Alliance. We are looking forward to receiving your revised manuscript.

Sincerely,

B. MANUSCRIPT ORGANIZATION AND FORMATTING:

November 29, 2022

RE: Life Science Alliance Manuscript #LSA-2022-01826-TR

Prof. Florian Heyd
Freie Universität Berlin
BCP
Takustr. 6
Berlin 14195
Germany

Dear Dr. Heyd,

Thank you for submitting your revised manuscript entitled "Branch point strength controls species-specific CamK2b alternative splicing and regulates LTP". We would be happy to publish your paper in Life Science Alliance pending final revisions necessary to meet our formatting guidelines.

- please add ORCID ID for corresponding author-you should have received instructions on how to do so
- please add the Twitter handle of your host institute/organization as well as your own or/and one of the authors in our system
- please consult our manuscript preparation guidelines <https://www.life-science-alliance.org/manuscript-prep> and make sure your manuscript sections are in the correct order
- please add your supplementary figure legends to the main manuscript text
- for Figure S3, please remove the panel A in the figure itself and in the figure callout; since this is the only panel for the figure, we do not need it designated
- please double-check your figure callouts for Figure S7; you have a callout for Figure S7E, but this is not in the figure or the legend
- please add a Data Availability Statement in the Materials and Methods section to repeat the accession info for the RNA-seq data. This GEO entry should also be made publicly accessible at this point.
- please add sizes next to all blots

A. FINAL FILES:

B. MANUSCRIPT ORGANIZATION AND FORMATTING:

Sincerely,

December 2, 2022

RE: Life Science Alliance Manuscript #LSA-2022-01826-TRR

Prof. Florian Heyd
Freie Universität Berlin
BCP
Takustr. 6
Berlin 14195
Germany

Dear Dr. Heyd,

Thank you for submitting your Research Article entitled "Branch point strength controls species-specific CamK2b alternative splicing and regulates LTP". It is a pleasure to let you know that your manuscript is now accepted for publication in Life Science Alliance. Congratulations on this interesting work.

DISTRIBUTION OF MATERIALS:

Again, congratulations on a very nice paper. I hope you found the review process to be constructive and are pleased with how the manuscript was handled editorially. We look forward to future exciting submissions from your lab.

Sincerely,
